# DOMAIN-INDEXING VARIATIONAL BAYES: INTERPRETABLE DOMAIN INDEX FOR DOMAIN ADAPTATION

**Zihao Xu[1*], Guang-Yuan Hao[2*], Hao He[3], Hao Wang[1]**

[1]Rutgers University, [2]Hong Kong University of Science and Technology,
[3]Massachusetts Institute of Technology,
zihao.xu@rutgers.edu, guangyuanhao@outlook.com
haohe@mit.edu, hw488@cs.rutgers.edu

## ABSTRACT

Previous studies have shown that leveraging *domain index* can significantly boost domain adaptation performance (Wang et al., 2020; Xu et al., 2022). However, such domain indices are not always available. To address this challenge, we first provide a formal definition of domain index from the probabilistic perspective, and then propose an adversarial variational Bayesian framework that infers domain indices from multi-domain data, thereby providing additional insight on domain relations and improving domain adaptation performance. Our theoretical analysis shows that our adversarial variational Bayesian framework finds the optimal domain index at equilibrium. Empirical results on both synthetic and real data verify that our model can produce interpretable domain indices which enable us to achieve superior performance compared to state-of-the-art domain adaptation methods. Code is available at https://github.com/Wang-ML-Lab/VDI.

## 1 INTRODUCTION

In machine learning, it is standard to assume that training data and test data share an identical distribution. However, this assumption is often violated (Ganin & Lempitsky, 2015; Romera et al., 2019; Sun et al., 2017; Yuan et al., 2019; Ramponi & Plank, 2020) when training and test data come from different domains. Domain adaptation (DA) tries to solve such a cross-domain generalization problem by producing domain-invariant features. Typically, DA methods enforce independence between a data point's latent representation and its *domain identity*, which is a *one-hot* vector indicating which domain the data point comes from (Ganin et al., 2016; Tzeng et al., 2017; Zhao et al., 2017; Zhang et al., 2019).

More recent studies have found that using *domain index*, which is a *real-value* scalar (or vector) to embed domain semantics, as a replacement of domain identity, significantly boosted domain adaptation performance (Wang et al., 2020; Xu et al., 2022). For instance, Wang et al. (2020) adapted sleeping stage prediction models across patients with different ages, with "age" as the domain index, and achieved superior performance compared to traditional models that split patients into groups by age and used discrete group IDs as domain identities (more discussion in Sec. J).

Although significant progress has been made in leveraging domain indices to improve domain adaptation (Wang et al., 2020; Xu et al., 2022), a major challenge exists: domain indices are not always available. This severely limits the applicability of such indexed DA methods. Thus a natural question is motivated: Can one infer the domain index as a latent variable from data?

This prompts us to first develop an expressive and formal definition of "domain index". We argue that an effective "domain index" (1) is independent of the data's encoding, (2) retains as much information on the data as possible, and (3) maximizes adaptation performance, e.g., accuracy (see Sec. 3.2 for rigorous descriptions). With this definition, we then develop an adversarial variational Bayesian deep learning model (Wang et al., 2015; Wang & Yeung, 2016; 2020) that describes intuitive conditional dependencies among the input data, labels, encodings, and the associated domain indices. Our theoretical analysis shows that maximizing our model's evidence lower bound while adversarially

---

[*]These authors contributed equally to this work.

training an additional discriminator (Ganin et al., 2016; Wang et al., 2020) is equivalent to inferring the optimal domain indices (according to our definition) that maximize the mutual information among the input data, labels, encodings, and the associated domain indices while minimizing the mutual information between the data's encodings and the domain indices. Our contributions are as follows:

- We identify the problem of inferring domain indices as latent variables, provide a rigorous definition of "domain index", and develop the first general method, dubbed variational domain indexing (VDI), for inferring such domain indices.
- Our theoretical analysis shows that training with VDI's final objective function is equivalent to inferring the optimal domain indices according to our definition.
- Experiments on both synthetic and real-world datasets show that VDI can infer non-trivial domain indices, thereby significantly improving performance over state-of-the-art DA methods.

## 2 RELATED WORK

**Typical Domain Adaptation.** There is a rich literature on domain adaptation (Pan & Yang, 2009; Pan et al., 2010; Long et al., 2018; Saito et al., 2018; Sankaranarayanan et al., 2018; Peng et al., 2019; Prabhu et al., 2021; Wang et al., 2020; Xu et al., 2022). Typically they try to align source-domain and target-domain data in the latent space, with the hope that such domain-invariant encodings can generalize well on unseen data. There are multiple ways to achieve such alignment, including distribution matching (Pan et al., 2010; Tzeng et al., 2014; Sun & Saenko, 2016; Peng et al., 2019; Nguyen-Meidine et al., 2021), self-training (Zou et al., 2018; Kumar et al., 2020; Prabhu et al., 2021), domain-specific normalization (Maria Carlucci et al., 2017; Mancini et al., 2019; Tasar et al., 2020), and deep learning models with adversarial training (Ganin et al., 2016; Tzeng et al., 2017; Zhang et al., 2019; Zhao et al., 2017; Chen et al., 2019; Dai et al., 2019). Most of these methods rely on (one-hot) *domain identities* for feature alignment.

**Domain Adaptation with Domain Identities and Domain Indices.** There are also works that generate domain identities from data to improve domain adaptation. Chen & Chao (2021) generates a sequence of domain identities for intermediate domains between a source domain and a target domain, trying to facilitate better incremental domain adaptation (Bobu et al., 2018). Deecke et al. (2021) generates a set of domain identities to split a dataset into different domains and perform multi-domain learning. Du et al. (2021); Lu et al. (2022) partition time series data by learning domain identities that maximize the domain-wise distribution gap. All these works focus on generating (ordinal or one-hot) domain identities. In contrast, our VDI assumes such domain identities are *already given* and focuses on inferring (continuous) domain indices, which contain richer and more interpretable information. Note that in our setting, since domain identities are *given*, methods such as Deecke et al. (2021) are equivalent to typical domain adaptation methods (Ganin et al., 2016; Tzeng et al., 2017; Prabhu et al., 2021), which are considered as our baselines in Sec. 5.

Recent studies have found that replacing (one-hot) domain identities with (continuous) domain indices improves adaptation performance (Wang et al., 2020; Xu et al., 2022). However, none of these works provide a canonical definition of "domain index"; instead they mainly rely on intuition, e.g., using rotation angles as domain indices for RotatingMNIST (Wang et al., 2020) and using graph node embeddings as domain indices for adaptation across graph-relational domains (Xu et al., 2022). More importantly, they assume that such domain indices are always available, which may not be true (Matsuura & Harada, 2020; Rebuffi et al., 2017); hence they are not applicable to our setting. Also related to our work is Peng et al. (2020a), which generates embeddings of visual domains to represent domain similarities; however, they do not formally define "domain embedding" and only work for classification tasks. In contrast, our VDI provides a rigorous and formal definition for "domain indices" and handles both classification and regression tasks.

## 3 METHOD

In this section, we formalize the definition of "domain index" and describe our VDI for inferring domain indices. We provide theoretical guarantees that VDI infers optimal domain indices in Sec. 4.

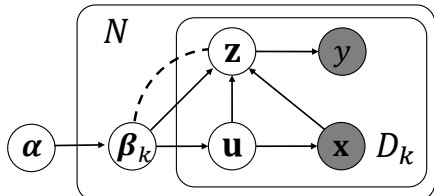 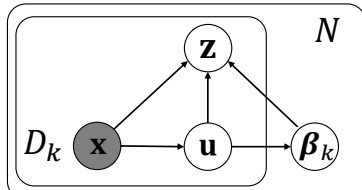

Figure 1: **Left:** Probabilistic graphical model for VDI's generative model. We introduce a new edge type, "---", to denote independence. $\boldsymbol{\beta}_k$ --- $\mathbf{z}$ enforces independence between $\mathbf{z}$ and $\boldsymbol{\beta}_k$, i.e, $p(\mathbf{z}|\boldsymbol{\beta}_k) = p(\mathbf{z})$ (see Appendix Sec. I for detailed discussion). **Right:** Probabilistic graphical model for the VDI's inference model.

## 3.1 PROBLEM SETTING AND NOTATION

We consider the unsupervised domain adaptation setting with $N$ domains in total. Each domain has domain identity $k \in \mathcal{K} = [N] \triangleq \{1, \ldots, N\}$; $k$ is in either the source domain identity set $\mathcal{K}_s$ or the target domain identity set $\mathcal{K}_t$. Each domain $k$ has $D_k$ data points. Given $n$ labeled data points $\{(\mathbf{x}_i^s, y_i^s, k_i^s)\}_{i=1}^n$ from source domains ($k_i^s \in \mathcal{K}_s$), and $m$ unlabeled data points $\{\mathbf{x}_i^t, k_i^t\}_{i=1}^m$ from target domains ($k_i^t \in \mathcal{K}_t$), we want to (1) predict the label $\{y_i^t\}_{i=1}^m$ for target domain data, and (2) infer global domain indices $\boldsymbol{\beta}_k \in \mathbb{R}^{B_\beta}$ for *each domain* and local domain indices $\mathbf{u}_i \in \mathbb{R}^{B_u}$ for *each data point*. $\boldsymbol{\alpha} = \{\boldsymbol{\mu}_\alpha, \boldsymbol{\sigma}_\alpha\}$ are the hyper-parameters for $\{\boldsymbol{\beta}_k\}_{k=1}^N$'s prior distributions. Note that each domain has only one global domain index, but has multiple local domain indices, one for each data point in the domain (more details in Sec. 3.3). We denote as $\mathbf{z} \in \mathbb{R}^{B_z}$ the data encoding generated from an encoder that takes $\mathbf{x}$ as input. We use $I(\cdot; \cdot)$ to denote mutual information.

## 3.2 FORMAL DEFINITION OF DOMAIN INDEX

We formally define "domain index" as follows (please refer to notations in Sec. 3.1 if needed):

**Definition 3.1** (**Domain Index**). *Given data $\mathbf{x}$ and label $y$, a domain-level variable $\boldsymbol{\beta}$ and a data-level variable $\mathbf{u}$ are called global and local domain indices, respectively, if there exists a data encoding $\mathbf{z}$ such that the following holds:*

*(1)* ***Independence between $\boldsymbol{\beta}$ and $\mathbf{z}$****: Global domain index $\boldsymbol{\beta}$ is independent of data encoding $\mathbf{z}$, i.e., $\boldsymbol{\beta} \perp\!\!\!\perp \mathbf{z}$, or equivalently $I(\boldsymbol{\beta}; \mathbf{z}) = 0$. This is to encourage domain-invariant data encoding $\mathbf{z}$.*

*(2)* ***Information Preservation of $\mathbf{x}$****: Data encoding $\mathbf{z}$, local domain index $\mathbf{u}$, and global domain index $\boldsymbol{\beta}$ preserves as much information on $\mathbf{x}$ as possible, i.e., maximizing $I(\mathbf{x}; \mathbf{u}, \boldsymbol{\beta}, \mathbf{z})$. This is to prevent $\boldsymbol{\beta}$ and $\mathbf{u}$ from collapsing to trivial solutions.*

*(3)* ***Label Sensitivity of $\mathbf{z}$****: The data encoding $\mathbf{z}$ should contain as much information on the label $y$ as possible to maximize prediction power, i.e., maximizing $I(y; \mathbf{z})$ conditioned on $\mathbf{z} \perp\!\!\!\perp \boldsymbol{\beta}$. This is to make sure the previous two constraints on $\boldsymbol{\beta}$, $\mathbf{u}$, and $\mathbf{z}$ do not harm prediction performance.*

*To summarize, $\boldsymbol{\beta}$ and $\mathbf{u}$ are considered the global and local domain indices, respectively, if $(\boldsymbol{\beta}, \mathbf{u}) = \text{argmax}_{\boldsymbol{\beta}, \mathbf{u}} I(\mathbf{x}; \mathbf{u}, \boldsymbol{\beta}, \mathbf{z}) + I(y; \mathbf{z})$ s.t. $I(\boldsymbol{\beta}; \mathbf{z}) = 0$.*

Later in Sec. 4, our theoretical analysis shows that maximizing our model's evidence lower bound while adversarially training an additional discriminator (Sec. 3.4) is equivalent to inferring the optimal domain indices according Definition 3.1. In Appendix A, we provides a rigorous discussion on the definition of "domain index".

## 3.3 DOMAIN-INDEXING VARIATIONAL BAYES FOR DOMAIN ADAPTATION

**Generative Process and Probabilistic Graphical Model.** Based on our definition, we propose our model: Variational Domain Indexing (VDI). The basic idea is to infer the domain indices as latent variables during domain adaptation. VDI is a generative model assuming the following generative process (see the corresponding graphical model in Fig. 1(left)). For each domain $k$,

(1) Draw global domain index $\boldsymbol{\beta}_k$ from the Gaussian distribution $p_\theta(\boldsymbol{\beta}|\boldsymbol{\alpha})$.
(2) For each data point $i$ with domain identity $k$:
    (a) Draw local domain index $\mathbf{u}_i$ from the Gaussian distribution $p_\theta(\mathbf{u}_i|\boldsymbol{\beta}_k)$.
    (b) Draw input $\mathbf{x}_i$ from the Gaussian distribution $p_\theta(\mathbf{x}_i|\mathbf{u}_i)$.

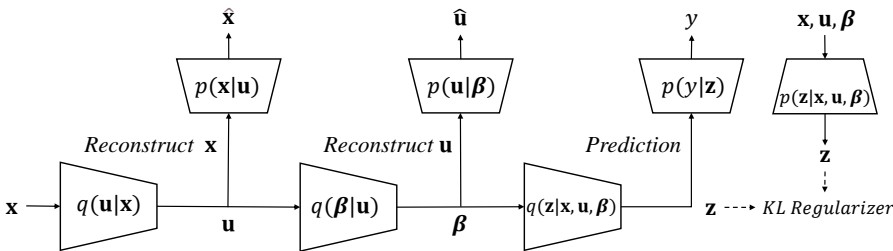

Figure 2: Network structure. For clarity, we omit subscripts of $q_\phi$ and $p_\theta$ as well as $p_\theta(\mathbf{z}|\mathbf{x}, \mathbf{u}, \boldsymbol{\beta})$'s input $(\mathbf{x}, \mathbf{u})$.

    (c) Draw data encoding $\mathbf{z}_i$ from the Gaussian distribution $p_\theta(\mathbf{z}_i|\mathbf{x}_i, \mathbf{u}_k, \boldsymbol{\beta}_i)$.

    (d) Draw label $y_i$ from the distribution $p_\theta(y_i|\mathbf{z}_i)$.

Besides typical conditional dependencies defined in the graphical model (Fig. 1(left)), we enforce additional independence between $\boldsymbol{\beta}$ and $\mathbf{z}$; such independence is represented as a dashed line "---" in Fig. 1(left). Note that there are multiple ways to satisfy such constraints during learning, e.g., using adversarial methods (Ganin et al., 2016; Tzeng et al., 2017; Zhang et al., 2019; Wang et al., 2020; Xu et al., 2022) and using the concentration loss (Xiao et al., 2021).

**Generative Model and Inference Model.** Based on Fig. 1(left), we factorize the generative model $p_\theta(\mathbf{x}, \mathbf{u}, \boldsymbol{\beta}, \mathbf{z}, y|\boldsymbol{\alpha})$ into five conditional distributions (omitting the subscript $i$ for clarity below):

$$p_\theta(\mathbf{x}, \mathbf{u}, \boldsymbol{\beta}, \mathbf{z}, y|\boldsymbol{\alpha}) = p_\theta(\boldsymbol{\beta}|\boldsymbol{\alpha})p_\theta(\mathbf{u}|\boldsymbol{\beta})p_\theta(\mathbf{x}|\mathbf{u})p_\theta(\mathbf{z}|\mathbf{x}, \mathbf{u}, \boldsymbol{\beta})p_\theta(y|\mathbf{z}), \qquad (1)$$

where $\boldsymbol{\theta}$ denotes the collection of parameters for the generative model, and $p_\theta(\boldsymbol{\beta}|\boldsymbol{\alpha}) = \mathcal{N}(\boldsymbol{\mu}_\alpha, \boldsymbol{\sigma}_\alpha^2)$ is a Gaussian distribution. The predictor $p_\theta(y|\mathbf{z})$ is a categorical distribution $Cat(f_y(\mathbf{z}; \boldsymbol{\theta}))$ for classification tasks and a Gaussian distribution $\mathcal{N}(\mu_y(\mathbf{z}; \boldsymbol{\theta}), \sigma_y^2(\mathbf{z}; \boldsymbol{\theta}))$ for regression tasks; here $f_y(\mathbf{z}; \boldsymbol{\theta})$, $\mu_y(\mathbf{z}; \boldsymbol{\theta})$, and $\sigma_y(\mathbf{z}; \boldsymbol{\theta})$ are neural networks taking $\mathbf{z}$ as input. Similarly, we have

$$p_\theta(\mathbf{u}|\boldsymbol{\beta}) = \mathcal{N}(\mu_u(\boldsymbol{\beta}; \boldsymbol{\theta}), \sigma_u^2(\boldsymbol{\beta}; \boldsymbol{\theta})), \qquad (2)$$

$$p_\theta(\mathbf{x}|\mathbf{u}) = \mathcal{N}(\mu_x(\mathbf{u}; \boldsymbol{\theta}), \sigma_x^2(\mathbf{u}; \boldsymbol{\theta})), \qquad (3)$$

$$p_\theta(\mathbf{z}|\mathbf{x}, \mathbf{u}, \boldsymbol{\beta}) = \mathcal{N}(\mu_z(\mathbf{x}, \mathbf{u}, \boldsymbol{\beta}; \boldsymbol{\theta}), \sigma_z^2(\mathbf{x}, \mathbf{u}, \boldsymbol{\beta}; \boldsymbol{\theta})). \qquad (4)$$

We use an inference model $q_\phi(\mathbf{u}, \boldsymbol{\beta}, \mathbf{z}|\mathbf{x})$ to approximate the posterior distributions of the latent variables, i.e., $p_\theta(\mathbf{u}, \boldsymbol{\beta}, \mathbf{z}|\mathbf{x})$. As shown in Fig. 1(right), we factorize $q_\phi(\mathbf{u}, \boldsymbol{\beta}, \mathbf{z}|\mathbf{x})$ as

$$q_\phi(\mathbf{u}, \boldsymbol{\beta}, \mathbf{z}|\mathbf{x}) = q_\phi(\mathbf{u}|\mathbf{x})q_\phi(\boldsymbol{\beta}|\mathbf{u})q_\phi(\mathbf{z}|\mathbf{x}, \mathbf{u}, \boldsymbol{\beta}), \qquad (5)$$

where $\phi$ denotes the collection of parameters for the inference model. Specifically, we have

$$q_\phi(\mathbf{u}|\mathbf{x}) = \mathcal{N}(\mu_u(\mathbf{x}; \phi), \sigma_u^2(\mathbf{x}; \phi)), \qquad (6)$$

$$q_\phi(\boldsymbol{\beta}|\mathbf{u}) = \mathcal{N}(\mu_\beta(\mathbf{u}; \phi), \sigma_\beta^2(\mathbf{u}; \phi)), \qquad (7)$$

$$q_\phi(\mathbf{z}|\mathbf{x}, \mathbf{u}, \boldsymbol{\beta}) = \mathcal{N}(\mu_z(\mathbf{x}, \mathbf{u}, \boldsymbol{\beta}; \phi), \sigma_z^2(\mathbf{x}, \mathbf{u}, \boldsymbol{\beta}; \phi)). \qquad (8)$$

Note that $\mu_.(\cdot; \cdot)$ and $\sigma_.(\cdot; \cdot)$ denote neural networks; $\boldsymbol{\theta}$, $\phi$ are neural network parameters. Here $q_\phi(\boldsymbol{\beta}|\mathbf{u})$ requires special treatment and will be discussed in Eqn. (14-17) below.

### 3.4 OBJECTIVE FUNCTION

**Evidence Lower Bound.** We use an evidence lower bound (ELBO) as an objective to learn the generative and inference models. Maximizing the ELBO learns the optimal variational distribution $q_\phi(\mathbf{u}, \boldsymbol{\beta}, \mathbf{z}|\mathbf{x})$ that best approximates the posterior distribution of the latent variables (including the domain indices) $p_\theta(\mathbf{u}, \boldsymbol{\beta}, \mathbf{z}|\mathbf{x})$. Specifically, we have the ELBO as:

$$\mathcal{L}_{ELBO}(\mathbf{x}, y) = \mathbb{E}_{q_\phi(\mathbf{u}, \boldsymbol{\beta}, \mathbf{z}|\mathbf{x})}[p_\theta(\mathbf{x}, \mathbf{u}, \boldsymbol{\beta}, \mathbf{z}, y|\boldsymbol{\alpha})] - \mathbb{E}_{q_\phi(\mathbf{u}, \boldsymbol{\beta}, \mathbf{z}|\mathbf{x})}[q_\phi(\mathbf{u}, \boldsymbol{\beta}, \mathbf{z}|\mathbf{x})]. \qquad (9)$$

With the factorization in Eqn. 1 and Eqn. 5, we decompose the ELBO as (omitting $\boldsymbol{\alpha}$ to avoid clutter):

$$\mathcal{L}_{ELBO}(\mathbf{x}, y) = \mathbb{E}_{q_\phi(\mathbf{u}|\mathbf{x})}[\log p_\theta(\mathbf{x}|\mathbf{u})] \qquad (10)$$

$$+ \mathbb{E}_{q_\phi(\mathbf{u}, \boldsymbol{\beta}, \mathbf{z}|\mathbf{x})}[\log p_\theta(y|\mathbf{z})] \qquad (11)$$

$$+ \mathbb{E}_{q_\phi(\mathbf{u}|\mathbf{x})}\mathbb{E}_{q_\phi(\boldsymbol{\beta}|\mathbf{u})}[\log p_\theta(\mathbf{u}|\boldsymbol{\beta})] \qquad (12)$$

$$- \mathbb{E}_{q_\phi(\mathbf{u}, \boldsymbol{\beta}, \mathbf{z}|\mathbf{x})}[KL[q_\phi(\boldsymbol{\beta}|\mathbf{u})||p_\theta(\boldsymbol{\beta})]] - KL[q_\phi(\mathbf{z}|\mathbf{x}, \mathbf{u}, \boldsymbol{\beta})||p_\theta(\mathbf{z}|\mathbf{x}, \mathbf{u}, \boldsymbol{\beta})] - \mathbb{E}_{q_\phi(\mathbf{u}|\mathbf{x})}[\log q_\phi(\mathbf{u}|\mathbf{x})], \qquad (13)$$

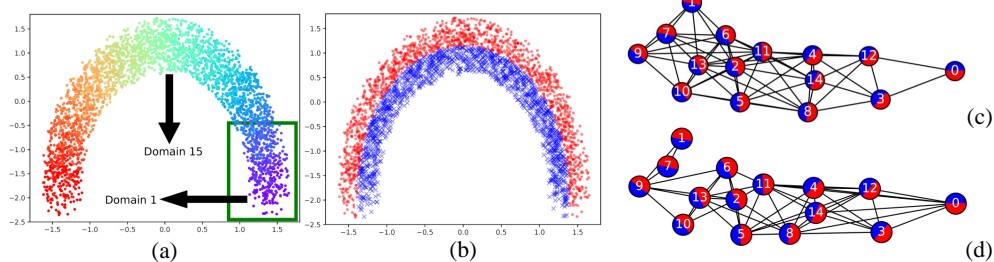

Figure 3: **(a)** The *Circle* dataset (Wang et al., 2020) with 30 domains, with different colors indicating ground-truth domain indices. The first 6 domains (in the green box) are source domains. **(b)** Ground-truth labels for *Circle*, with red dots and blue crosses as positive and negative data points, respectively. **(c)** *Ground-truth* domain graph for *DG-15*. We use 'red' and 'blue' to roughly indicate positive and negative data points in a domain. **(d)** VDI's *inferred* domain graph for *DG-15*, with an AUC of 0.83.

Each term above is computable with our neural network parameterization (see the network structure in Fig. 2); for target domains Eqn. 11 is excluded. Below, we describe each term's intuition.

(1) **Reconstruct Data x from u (Eqn. 10).** $q_\phi(\mathbf{u}|\mathbf{x})$ and $p_\theta(\mathbf{x}|\mathbf{u})$ aim to reconstruct data $\mathbf{x}$ using the inferred $\mathbf{u}$, encouraging $\mathbf{u}$ to preserve as much information on $\mathbf{x}$ as possible.

(2) **Predict Label $y$ from Latent $z$ (Eqn. 11).** This term samples $\mathbf{u}$, $\boldsymbol{\beta}$, and $\mathbf{z}$ from $q_\phi(\mathbf{u}|\mathbf{x})$, $q_\phi(\boldsymbol{\beta}|\mathbf{u})$ and $q_\phi(\mathbf{z}|\mathbf{x}, \mathbf{u}, \boldsymbol{\beta})$, respectively, and then uses $\mathbf{z}$ to predict $y$ in $p_\theta(y|\mathbf{z})$, encouraging $\mathbf{z}$ to contain as much information on $y$ as possible to maximize prediction performance.

(3) **Reconstruct Local Domain Index u from $\boldsymbol{\beta}$ (Eqn. 12).** Eqn. 12 samples $\mathbf{u}$ and $\boldsymbol{\beta}$ from $q_\phi(\mathbf{u}|\mathbf{x})$ and $q_\phi(\boldsymbol{\beta}|\mathbf{u})$, respectively, and then uses the inferred $\boldsymbol{\beta}$ to reconstruct local domain index $\mathbf{u}$ in $p_\theta(\mathbf{u}|\boldsymbol{\beta})$, encouraging $\boldsymbol{\beta}$ to preserve as much information on $\mathbf{u}$ and $\mathbf{x}$ as possible.

(4) **Regularize All Latent Variables $\mathbf{u}, \boldsymbol{\beta}, \mathbf{z}$ (Eqn. 13).** Eqn. 13 includes two KL divergence terms between the inference model $q_\phi(\cdot)$ and the generative model $p_\theta(\cdot)$ as well as an entropy term for $q_\phi(\mathbf{u}|\mathbf{x})$; they all serve as regularizers to prevent overfitting. For example, the first regularization term implies that $q(\boldsymbol{\beta}|\mathbf{u})$ should be close to the prior distribution $p(\boldsymbol{\beta})$.

**Global Domain Index $\boldsymbol{\beta}$ and Local Domain Index u.** VDI uses a bi-level structure for domain indices: local domain index $\mathbf{u} \in \mathbb{R}^{B_u}$ and global domain index $\boldsymbol{\beta} \in \mathbb{R}^{B_\beta}$. Both $\mathbf{u}$ and $\boldsymbol{\beta}$ are low-dimensional compared to $\mathbf{x} \in \mathbb{R}^{B_x}$, i.e., $B_u \ll B_x$ and $B_\beta \ll B_x$. The local domain index $\mathbf{u}$ is a low-dimensional vector (e.g., $B_u = 4$) containing domain information for high-dimensional data $\mathbf{x}$ (e.g., an image with $B_x = 256 \times 256$). The global domain index $\boldsymbol{\beta}$ is an aggregation of all local domain indices $\mathbf{u}$ for data from the same domain. Note that different data points $\mathbf{x}_i$ and $\mathbf{x}_j$ in the same domain ($k_i = k_j$) have *different local domain indices*, i.e., $\mathbf{u}_i \neq \mathbf{u}_j$, but *share the same global domain index*, i.e., $\boldsymbol{\beta}_{k_i} = \boldsymbol{\beta}_{k_j}$. VDI's final goal is to infer the optimal global domain indices $\{\boldsymbol{\beta}_k\}_{k=1}^N$ given only the data $(\mathbf{x}_i, y_i)$ and domain identities $k_i$, thereby providing better interpretability and domain adaptation performance. See Sec. K for more discussion on $\boldsymbol{\beta}$ and $\mathbf{u}$.

**Difference between Domain Identities $k$ and Global Domain Indices $\boldsymbol{\beta}$.** Note that domain identities $k$ are discrete values and therefore cannot describe rich relations (e.g., similarity and distance) among domains. In contrast, global domain indices $\boldsymbol{\beta}$ are continuous vectors and therefore contain much richer information that describes relations (e.g., similarity and distance) among domains (see Sec. 5 for empirical results). Our VDI assumes $k$ is available and tries to infer $\boldsymbol{\beta}$.

**Inferring Global Domain Indices $q_\phi(\boldsymbol{\beta}|\mathbf{u})$ in Eqn. 5.** For each domain $k$, global domain index $\boldsymbol{\beta}_k$ should aggregate domain information of all data in this domain. We therefore propose to leverage local domain indices of all domain $k$'s data points, $\mathbf{U}_k = [\mathbf{u}_i]_{k_i=k} \in \mathbb{R}^{D_k \times B_u}$, to infer the global domain index $\boldsymbol{\beta}_k$. Specifically, our process consists of four steps: (1) **Grouping $\mathbf{u}_i$ in Domain $k$.** Group all local domain indices from the same domain $k$ into one local index matrix (set), i.e., $\mathbf{U}_k = [\mathbf{u}_i]_{k_i=k} \in \mathbb{R}^{D_k \times B_u}$. (2) **Pairwise Domain Distance.** Calculate the Earth Mover's distance (EMD) (Rubner et al., 2000) between each pair of local index matrices (sets) $\mathbf{S}_{k,j} = \mathbf{f}_{EMD}(\mathbf{U}_k, \mathbf{U}_j) \in \mathbb{R}^{N \times N}$, where $\mathbf{S}_{k,j}$ is the EMD between domain $k$ and $j$. (3) **Raw Global Domain Indices.** According to the pairwise domain distance matrix $\mathbf{S}$, use multi-dimensional scaling (MDS) (Borg & Groenen, 2005) to map each domain $k$ into a $B_\beta$-dimensional space and obtain the raw global domain index $\boldsymbol{\beta}_k^r \in \mathbb{R}^{B_\beta}$, i.e., $[\boldsymbol{\beta}_k^r]_{k=1}^N = \mathbf{f}_{MDS}(\mathbf{S}) = [\mathbf{f}_{MDS}^k(\mathbf{S})]_{k=1}^N$. (4) **Final Global Domain Indices.** Feed the raw index $\boldsymbol{\beta}_k^r$ into the inference neural network to obtain the variational distribution $\mathcal{N}\left(\mu_r(\boldsymbol{\beta}_k^r; \boldsymbol{\phi}), \sigma_r^2(\boldsymbol{\beta}_k^r; \boldsymbol{\phi})\right)$ for the final global domain index $\boldsymbol{\beta}_k \in \mathbb{R}^{B_\beta}$, where $\boldsymbol{\phi}$ is the

inference network parameters. We summarize these four steps below:

$$\text{Grouping } \mathbf{u}_i \text{ in Domain } k: \quad \mathbf{U}_k = [\mathbf{u}_i]_{k_i=k} \in \mathbb{R}^{D_k \times B_u}, \tag{14}$$

$$\text{Pairwise Domain Distance: } \quad \mathbf{S} = [\mathbf{f}_{EMD}(\mathbf{U}_k, \mathbf{U}_j)]_{k=1,j=1}^{N,N} \in \mathbb{R}^{N \times N}, \tag{15}$$

$$\text{Raw Global Domain Indices: } \quad \boldsymbol{\beta}_k^r = \mathbf{f}_{MDS}^k(\mathbf{S}) \in \mathbb{R}^{B_\beta}, \tag{16}$$

$$\text{Final Global Domain Indices: } \quad \boldsymbol{\beta}_k \sim \mathcal{N}\big(\mu_r(\boldsymbol{\beta}_k^r), \sigma_r^2(\boldsymbol{\beta}_k^r); \boldsymbol{\phi}\big) \in \mathbb{R}^{B_\beta}. \tag{17}$$

**Discriminator with an Adversarial Loss.** To enforce independence between $\boldsymbol{\beta}$ and $\mathbf{z}$, i.e., Part (1) of Definition 3.1, we train an additional discriminator $D$ with an adversarial loss while maximizing the ELBO in Eqn. 9. The discriminator is a neural network $D(\cdot)$ that takes $\mathbf{z}$ as input and predicts the global domain index $\hat{\boldsymbol{\beta}}$ and domain identity $\hat{k}$. Essentially, $D(\cdot)$ plays a minimax game with the encoder inference network $q_\phi(\mathbf{z}|\mathbf{x}, \mathbf{u}, \boldsymbol{\beta})$: $D(\cdot)$ tries to reconstruct the global domain index $\hat{\boldsymbol{\beta}}$ and domain identity $\hat{k}$, while the encoder $q_\phi(\mathbf{z}|\mathbf{x}, \mathbf{u}, \boldsymbol{\beta})$ tries to prevent $D(\cdot)$ from doing so by generating domain-invariant encoding $\mathbf{z}$. Denoting as $R_D$ the reconstruction loss, the discriminator loss $\mathcal{L}_D$ can be written as:

$$\mathcal{L}_D = R_D(\boldsymbol{\beta}, \hat{\boldsymbol{\beta}}, k, \hat{k}) \tag{18}$$

In Sec. 4, we will prove that $\boldsymbol{\beta}$ is guaranteed to be independent of $\mathbf{z}$ if $k$ is independent of $\mathbf{z}$. We therefore simplify Eqn. 18 into only classifying the domain identity $k$ and use the log-likelihood as $\mathcal{L}_{D,\phi}$:

$$\mathcal{L}_{D,\phi} = \mathbb{E}_{p(k,\mathbf{x})}\mathbb{E}_{q_\phi(\mathbf{z}|\mathbf{x})}[\log D(k|\mathbf{z})] \tag{19}$$

**Final Objective Function.** Putting Eqn. 9 and Eqn. 19 together, we have our final objective function:

$$\max_{\theta,\phi} \min_D \mathcal{L}_{VDI} = \max_{\theta,\phi} \min_D \mathcal{L}_{\theta,\phi} - \lambda_d \mathcal{L}_{D,\phi}$$
$$= \max_{\theta,\phi} \min_D \mathbb{E}_{p(\mathbf{x},y)}[\mathcal{L}_{ELBO}(\mathbf{x}, y)] - \lambda_d \mathbb{E}_{p(k,\mathbf{x})}\mathbb{E}_{q_\phi(\mathbf{z}|\mathbf{x})}[\log D(k|\mathbf{z})], \tag{20}$$

where $\lambda_d$ is a hyper-parameter balancing two terms.

## 4 THEORY

Below we provide theoretical guarantees for VDI's objective function (Eqn. 20). We analyze the first term in Lemma 4.1 and the second term in Lemma 4.2, show that Eqn. 20 lower-bounds a combination of mutual information terms (including $I(y; \mathbf{z})$, $I(\mathbf{x}; \mathbf{u}, \boldsymbol{\beta}, \mathbf{z})$, and $I(\mathbf{z}; \boldsymbol{\beta})$) plus some constants in Theorem 4.1, and then show that one can learn domain indices $\boldsymbol{\beta}$ that satisfies Definition 3.1 when Eqn. 20's global optimum is achieved (Theorem 4.2). **All proofs are in Appendix B.**

We start by analyzing VDI's ELBO term $\mathcal{L}_{ELBO}(\mathbf{x}, y)$ in Eqn. 20 and proving that it is upper bounded by $I(y; \mathbf{z}) + I(\mathbf{x}; \mathbf{u}, \boldsymbol{\beta}, \mathbf{z})$ plus some constants in Lemma 4.1 below.

**Lemma 4.1 (Upper Bound of the ELBO of $p_\theta(\mathbf{x}, y)$).** *The ELBO of $p_\theta(\mathbf{x}, y)$ is upper bounded by the mutual information among observable variables $\mathbf{x}, y$ and latent variables $\mathbf{u}, \boldsymbol{\beta}, \mathbf{z}$ as below:*

$$\mathbb{E}_{p(\mathbf{x},y)}[\mathcal{L}_{ELBO}(p_\theta(\mathbf{x}, y))] \leq I(y; \mathbf{z}) + I(\mathbf{x}; \mathbf{u}, \boldsymbol{\beta}, \mathbf{z}) - [H(y) + H(\mathbf{x})]. \tag{21}$$

Since the entropy terms $H(y)$ and $H(\mathbf{x})$ in Eqn. 21 are both constant, maximizing the ELBO term $\mathcal{L}_{ELBO}(\mathbf{x}, y)$ in Eqn. 20 is equivalent to maximizing $I(\mathbf{x}; \mathbf{u}, \boldsymbol{\beta}, \mathbf{z})$ and $I(y; \mathbf{z})$, corresponding to Parts (2) and (3) of Definition 3.1, respectively.

Next we analyze VDI's adversarial term $\mathcal{L}_{D,\phi}$ of Eqn. 20 in Lemma 4.2 below.

**Lemma 4.2 (Information Decomposition of the Adversarial Loss).** *The global maximum of adversarial loss w.r.t. discriminator $D$ is decomposed as below:*

$$\max_D \mathbb{E}_{p(k,\mathbf{x})}\mathbb{E}_{q_\phi(\mathbf{z}|\mathbf{x})}[\log D(k|\mathbf{z})] = I(\mathbf{z}; \boldsymbol{\beta}) + I(\mathbf{z}; k|\boldsymbol{\beta}) - H(k), \tag{22}$$

*and the global minimum of $\max_D \mathbb{E}_{p(k,\mathbf{x})}\mathbb{E}_{q_\phi(\mathbf{z}|\mathbf{x})}[\log D(k|\mathbf{z})]$ is achieved if and only if $I(\mathbf{z}; \boldsymbol{\beta}) = I(\mathbf{z}; k|\boldsymbol{\beta}) = 0$.*

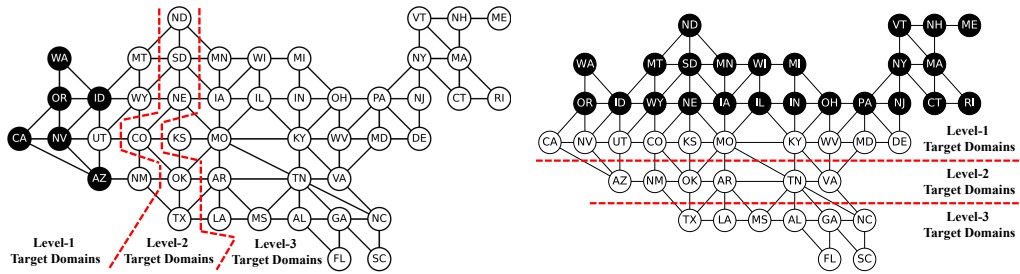

Figure 4: Domain graphs for two adaptation tasks on *TPT-48*; black nodes indicate source domains, and white nodes indicate target domains. **Left:** Adaptation from the 6 states in the west to the 42 states in the east. **Right:** Adaptation from the 24 states in the north to the 24 states in the south.

Lemma 4.2 above shows that one can decompose $\max_D \mathbb{E}_{p(k,\mathbf{x})}\mathbb{E}_{q_\phi(\mathbf{z}|\mathbf{x})}[\log D(k|\mathbf{z})]$ into several information theoretic terms, including $I(\mathbf{z};\boldsymbol{\beta})$, which is related to Part (1) of Definition 3.1.

With Lemma 4.1 and Lemma 4.2, we then show that VDI's objective function in Eqn. 20 lower-bounds a combination of mutual information terms plus some constant entropy terms in Theorem 4.1 below.

**Theorem 4.1** (**Objective Function as a Lower Bound**). *The objective function involves both the ELBO of $p_\theta(\mathbf{x}, y)$ and adversarial loss $\mathbb{E}_{p(k,\mathbf{x})}\mathbb{E}_{q_\phi(\mathbf{z}|\mathbf{x})}[\log D(k|\mathbf{z})]$, and it is the lower bound for a combination mutual information and entropy terms:*

$$\mathbb{E}_{p(\mathbf{x},y)}[\mathcal{L}_{ELBO}(\mathbf{x},y)] - \max_D \mathbb{E}_{p(k,\mathbf{x})}\mathbb{E}_{q_\phi(\mathbf{z}|\mathbf{x})}[\log D(k|\mathbf{z})] \tag{23}$$

$$\leq I(y;\mathbf{z}) + I(\mathbf{x};\mathbf{u},\boldsymbol{\beta},\mathbf{z}) - I(\mathbf{z};\boldsymbol{\beta}) - I(\mathbf{z};k|\boldsymbol{\beta}) - [H(y) + H(\mathbf{x}) - H(k)]. \tag{24}$$

With Theorem 4.1, we are now ready to analyze the global optimum of the minimax game in Eqn. 20.

**Theorem 4.2** (**Global Optimum of VDI**). *In VDI, when the global optimum (Eqn. 20) is achieved, it is guaranteed that (1) $I(\mathbf{z};\boldsymbol{\beta}) = 0$, (2) $I(\mathbf{x};\mathbf{u},\boldsymbol{\beta},\mathbf{z})$ is maximized, and (3) $I(y;\mathbf{z})$ is maximized.*

As Theorem 4.2 states, the global optimum of Eqn. 20 is guaranteed to satisfy all three conditions in Definition 3.1; therefore training VDI using the minimax game objective Eqn. 20 is equivalent to inferring the optimal domain indices.

## 5 EXPERIMENTS

In this section, we compare VDI with existing DA methods on both synthetic and real-world datasets.

### 5.1 DATASETS

***Circle*** (Wang et al., 2020) is a synthetic dataset with 30 domains for binary classification. Fig. 3(a) shows 30 domains of *Circle* in different colors. Fig. 3(b) shows positive (red) and negative (blue) data points, The first 6 domains are source domains, and the remaining 24 domains are target domains.

***DG-15 and DG-16*** (Xu et al., 2022). *DG-15* (Fig. 3(c)) and *DG-60* (Fig. 9(b)) are synthetic datasets with 15 and 60 domains for binary classification, respectively. In both datasets, we use 6 connected domains as the source domains and use others as target domains (see Table 4 in Sec. F for details).

***TPT-48*** (Xu et al., 2022) is a real-world regression dataset that contains monthly average temperature for the 48 contiguous states in the US from 2008 to 2019. We use the first 6 months' temperature as model input to predict the next 6 months' temperature. We formulate two DA tasks (Fig. 4):

- $W$ (6) $\rightarrow$ $E$ (42): Adapting models from the 6 states in the west to the 42 states in the east.
- $N$ (24) $\rightarrow$ $S$ (24): Adapting models from the 24 states in the north to the 24 states in the south.

We treat target domains one hop away from the closest source domain as *Level-1 Target Domains*, those two hops away as *Level-2 Target Domains*, and those more than two hops away as *Level-3 Target Domains* (see Fig. 4 for an illustration).

***CompCars*** (Yang et al., 2015) is a car image dataset with attributes including car types, viewpoints, and years of manufacture (YOMs). The task is to recognize the car type given an image. In *CompCars*, data with each view point and each YOM is treated as a single domain. We choose a subset of CompCars with 4 car types (MPV, SUV, sedan and hatchback), 5 viewpoints (front (F), rear

Table 1: Accuracy (%) on *Circle*, *DG-15* and *DG-60*.

| Method | Source-Only | DANN | ADDA | CDANN | MDD | SENTRY | D2V | VDI (Ours) |
|--------|-------------|------|------|-------|-----|--------|-----|------------|
| *Circle* | 55.5 | 53.4 | 56.2 | 54.9 | 53.4 | 59.5 | 60.1 | **94.3** |
| *DG-15* | 39.7 | 43.4 | 33.5 | 38.8 | 37.2 | 42.6 | 79.9 | **94.7** |
| *DG-60* | 55.0 | 66.3 | 60.8 | 65.3 | 54.6 | 51.3 | 82.1 | **95.9** |

Table 2: MSE for various DA methods for both tasks W (6) → E (42) and N (24) → S (24) on *TPT-48*. We report the average MSE of all domains as well as more detailed average MSE of Level-1, Level-2, Level-3 target domains, respectively (Fig. 4). D2V cannot perform regression and thus has no results. Note that there is only one single DA model per column. We mark the best result with **bold face**.

| Task | Domain | Source-Only | DANN | ADDA | CDANN | MDD | SENTRY | D2V | VDI (Ours) |
|------|--------|-------------|------|------|-------|-----|--------|-----|------------|
| W (6)→E (42) | Average of 4 Level-1 Domains | **1.184** | 1.984 | 5.448 | 6.168 | 5.544 | 2.512 | - | 2.160 |
| | Average of 6 Level-2 Domains | 3.128 | 5.112 | 7.624 | 7.016 | 7.912 | 5.136 | - | **3.000** |
| | Average of 32 Level-3 Domains | 5.272 | 5.880 | 7.256 | 6.968 | 8.008 | 5.872 | - | **2.448** |
| | Average of All 42 Domains | 4.576 | 5.400 | 7.136 | 6.896 | 7.76 | 5.456 | - | **2.496** |
| N (24)→S (24) | Average of 10 Level-1 Domains | 1.648 | 1.832 | 5.872 | 1.832 | 2.736 | 3.976 | - | **1.536** |
| | Average of 6 Level-2 Domains | 3.128 | 3.296 | 6.888 | 2.856 | 6.144 | 3.760 | - | **2.584** |
| | Average of 8 Level-3 Domains | 9.280 | 6.744 | 7.088 | 7.688 | 10.608 | **3.672** | - | 5.624 |
| | Average of All 24 Domains | 4.560 | 3.840 | 6.528 | 4.040 | 6.216 | 3.816 | - | **3.160** |

(R), side (S), front-side (FS), and rear-side (RS)), ranging from 2009 to 2014. It contains 30 domains (5 viewpoints × 6 YOMs) with 18735 images in total. We choose the domain with front view and YOM 2009 as the source domain, and all the others as target domains.

## 5.2 BASELINES

We compared our proposed VDI with state-of-the-art DA methods, including Domain Adversarial Neural Networks (**DANN**) (Ganin et al., 2016), Adversarial Discriminative Domain Adaptation (**ADDA**) (Tzeng et al., 2017), Conditional Domain Adaptation Neural Networks (**CDANN**) (Zhao et al., 2017), Margin Disparity Discrepancy (**MDD**) (Zhang et al., 2019), **SENTRY** (Prabhu et al., 2021) and Domain to Vector (**D2V**) (Peng et al., 2020b). We also report results when the model is only trained in the source domains without adapting to the target domains (**Source-Only**). Different from VDI that works for both classification and regression tasks, MDD, SENTRY and D2V only work for classification tasks. We managed to adapt MDD and SENTRY for the regression tasks on *TPT-48* (see App. H for details); D2V cannot be adapted and has no results for *TPT-48*. Both Wang et al. (2020) and Xu et al. (2022) assume domain indices are available; therefore they are not applicable to our settings where the goal is to infer domain indices (which are *unavailable* from data).

## 5.3 RESULTS

***Circle, DG-15 and DG-60***. Table 1 shows the accuracy of evaluated methods on *Circle*, *DG-15*, and *DG-60*; all datasets have complex domain relations, making it challenging to perform domain adaptation without knowing ground-truth domain indices. Indeed, we observe that on *Circle*, all baselines only perform marginally better than random guess (50% accuracy). Moreover, on *DG-15* most of baselines perform even worse than a random guess, possibly due to overfitting the source domains[1]. In contrast, our VDI achieves very high accuracy (over 94%) on all three datasets, significantly outperforming all baselines, thanks to the inferred indices (e.g., Fig. 3(d) and Fig. 7).

To verify that VDI infers non-trivial domain indices $\beta$, we connect the domain pairs within a distance threshold ($\|\beta_k - \beta_j\| < \epsilon$) to reconstruct the domain graph on *DG-15* and *DG-60*. Compared with the ground-truth domain graphs, VDI achieves area under the ROC curve (AUC) of 0.83 for *DG-15* and 0.91 for *DG-60*. Fig. 3(d) shows an example inferred domain graph for *DG-15*.

***TPT-48***. Table 2 shows the mean square error (MSE) for all the methods on *TPT-48*. In terms of average MSE across all domains, we observe that most methods suffer from negative transfer on both tasks, with only DANN and SENTRY marginally improving upon Source-Only. In contrast, our VDI can further improve the performance and achieve the lowest average MSE on both tasks.

---

[1] Note that different from (Wang et al., 2020; Xu et al., 2022), all evaluated methods only have access to domain identities $k$, but not ground-truth domain indices $\beta$, since the goal is to infer $\beta$ in our setting.

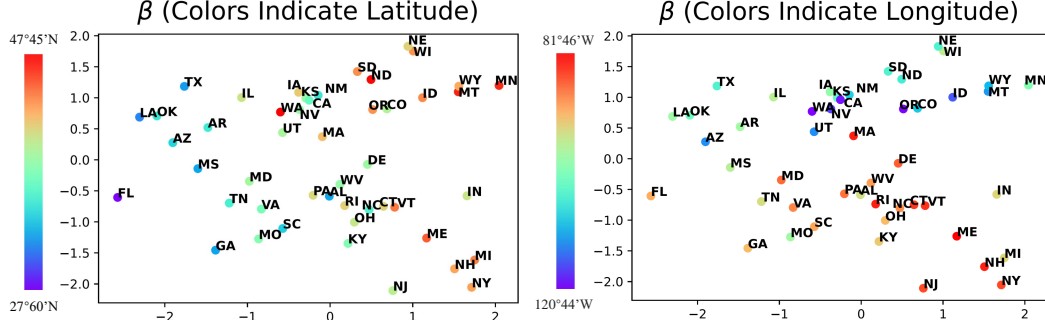

Figure 5: Inferred domain indices for 48 domains in *TPT-48*. We color inferred domain indices according to ground-truth indices, latitude (**left**) and longitude (**right**). VDI's inferred indices are correlated with true indices, even though *VDI does not have access to true indices during training*.

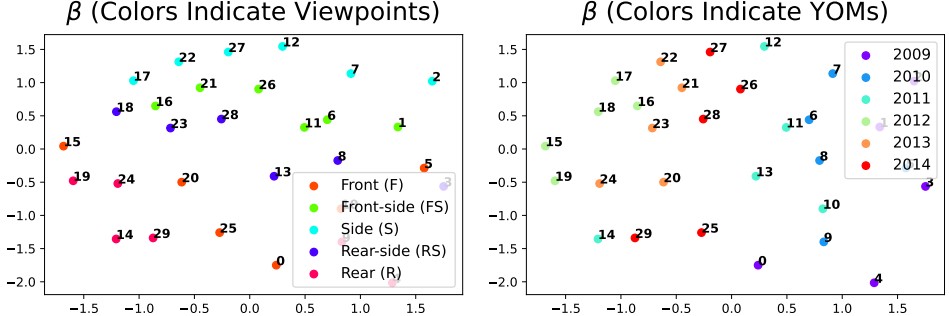

Figure 6: Inferred domain indices for 30 domains in *CompCars*. We color inferred domain indices according to ground-truth indices, viewpoints (**left**) and YOMs (**right**). Numbers on the circles are domain identities (discrete values). VDI's inferred indices are correlated with true indices, even though *VDI does not have access to true indices during training*. See larger figures in App. L.

Table 3: Accuracy (%) on CompCars (4-Way Classification).

| Method | Source-Only | DANN | ADDA | CDANN | MDD | SENTRY | D2V | VDI (Ours) |
|--------|-------------|------|------|-------|-----|--------|-----|------------|
| *CompCars* | 39.1 | 38.9 | 42.8 | 41.8 | 41.4 | 41.8 | 40.7 | **43.9** |

Fig. 5 plots the inferred domain indices $\boldsymbol{\beta} \in \mathbb{R}^2$ for all 48 domains. For reference, we color the inferred domain indices according to ground-truth latitude (Fig. 5(left)) and longitude (Fig. 5(right)); note that *VDI does not have access to latitude and longitude during training*. The plots show that VDI's inferred domain indices are highly correlated with each domain's latitude and longitude. For example, Florida (FL) has the lowest latitude among all 48 states and is hence the left-most circle in Fig. 5(left). We also observe that states with similar latitude or longitude do have similar domain indices $\boldsymbol{\beta}$. These results demonstrate that VDI can infer reasonable domain indices.

***CompCars.*** Table 3 shows the classification accuracy on all DA methods. Results show that most of the methods outperform Source-Only, with our VDI achieving the most significant improvement. Fig. 6 plots the inferred domain indices $\boldsymbol{\beta} \in \mathbb{R}^2$ for all 30 domains. For reference, we also color the plotted circles according to YOMs (Fig. 6(left)) and viewpoints (Fig. 6(right)); note that *VDI does not have access to YOMs and viewpoints during training*. Interestingly, we have the following observations that are consistent with intuition: (1) domains with the same viewpoint or YOM have similar domain indices; (2) domains with "front-side" and "rear-side" viewpoints have similar domain indices; (3) domains with "front" and "rear" viewpoints have similar domain indices.

## 6 CONCLUSION

We identify the problem of inferring domain indices as latent variables, provide a rigorous definition of "domain index", develop the first general method for addressing it, and provide detailed theoretical analysis as well as empirical results. We demonstrate the effectiveness of our proposed VDI for inferring domain indices and show its potential for significant practical applications. As a limitation, our method still assumes the availability of domain identities to identify different domains. Therefore it would be interesting future work to explore jointly inferring domain indices and domain identities.

## ACKNOWLEDGEMENT

The authors thank the reviewers/AC for the constructive comments to improve the paper. ZX and HW are partially supported by NSF Grant IIS-2127918. The views and conclusions contained herein are those of the authors and should not be interpreted as necessarily representing the official policies, either expressed or implied, of the sponsors.

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

## A  FORMAL DEFINITION OF DOMAIN INDEX

Given predefined variables of data $\mathbf{x}$, label $y$ and domain, let $\boldsymbol{\beta}$ be a domain associated variable, i.e., in each domain it has the same value, and let $\mathbf{u}$ be a data associated variable, i.e., for each data point it can have a different value.

**Definition A.1** (**Orthogonal Information**). *For any domain associated variable $\boldsymbol{\beta}$ and data associated variable $\mathbf{u}$, we define their orthogonal information as the following:*

*(1) Data orthogonal information $I_{\mathbf{x}}(\boldsymbol{\beta}, \mathbf{u})$: $I_{\mathbf{x}}(\boldsymbol{\beta}, \mathbf{u}) := \max_{\mathbf{z}:\mathbf{z} \perp\!\!\!\perp \boldsymbol{\beta}} I(\mathbf{x}; \mathbf{u}, \boldsymbol{\beta}, \mathbf{z})$.*

*(2) Label orthogonal information $I_{y}(\boldsymbol{\beta})$: $I_{y}(\boldsymbol{\beta}) := \max_{\mathbf{z}:\mathbf{z} \perp\!\!\!\perp \boldsymbol{\beta}} I(y; \mathbf{z})$.*

**Definition A.2** (**Domain Index**). *We call $(\boldsymbol{\beta}, \mathbf{u})$ domain index if they maximize the sum of data and label orthogonal information, i.e., $\boldsymbol{\beta}, \mathbf{u} \in \mathrm{argmax}_{\boldsymbol{\beta}', \mathbf{u}'} I_{\mathbf{x}}(\boldsymbol{\beta}', \mathbf{u}') + I_{y}(\boldsymbol{\beta}')$.*

**Lemma A.1.** *By Definition A.2, Domain Index maximizes both data orthogonal information and label orthogonal information, i.e., if $(\boldsymbol{\beta}, \mathbf{u})$ is domain index, then $I_{\mathbf{x}}(\boldsymbol{\beta}, \mathbf{u}) = H(\mathbf{x})$ and $I_{y}(\boldsymbol{\beta}) = I(y; \mathbf{x})$.*

*Proof.* We prove the lemma above by first constructing a trivial domain index. Let $\boldsymbol{\beta}_0, \mathbf{u}_0$ be variables that are independent of $\mathbf{x}$, i.e., $\boldsymbol{\beta}_0 \perp\!\!\!\perp \mathbf{x}$ and $\mathbf{u}_0 \perp\!\!\!\perp \mathbf{x}$; let $\mathbf{z}_0 = \mathbf{x}$ and hence $I(\mathbf{x}; \mathbf{z}_0) = H(\mathbf{x})$. We have $I_{\mathbf{x}}(\boldsymbol{\beta}_0, \mathbf{u}_0) \geq I(\mathbf{x}; \mathbf{u}_0, \boldsymbol{\beta}_0, \mathbf{z}_0) = H(\mathbf{x})$ and $I_{y}(\boldsymbol{\beta}_0) \geq I(y; \mathbf{z}_0) = I(y; \mathbf{x})$. For any domain index $(\boldsymbol{\beta}, \mathbf{u})$ by Definition A.2, we have $H(\mathbf{x}) + I(y; \mathbf{x}) \leq I_{\mathbf{x}}(\boldsymbol{\beta}_0, \mathbf{u}_0) + I_{y}(\boldsymbol{\beta}_0) \leq I_{\mathbf{x}}(\boldsymbol{\beta}, \mathbf{u}) + I_{y}(\boldsymbol{\beta}) \leq H(\mathbf{x}) + I(y; \mathbf{x})$; therefore $I_{\mathbf{x}}(\boldsymbol{\beta}, \mathbf{u}) + I_{y}(\boldsymbol{\beta}) = H(\mathbf{x}) + I(y; \mathbf{x})$. Since $I_{\mathbf{x}}(\boldsymbol{\beta}, \mathbf{u}) \leq H(\mathbf{x})$ and $I_{y}(\boldsymbol{\beta}) \leq I(y; \mathbf{x})$, we then have $I_{\mathbf{x}}(\boldsymbol{\beta}, \mathbf{u}) = H(\mathbf{x})$ and $I_{y}(\boldsymbol{\beta}) = I(y; \mathbf{x})$, concluding the proof. $\square$

# B THEORETICAL ANALYSIS

**Lemma B.1** (**Upper Bound of the ELBO of** $p_\theta(\mathbf{x}, y)$). *The ELBO of $p_\theta(\mathbf{x}, y)$ is upper bounded by the mutual information among observable variables $\mathbf{x}, y$ and latent variables $\mathbf{u}, \boldsymbol{\beta}, \mathbf{z}$ as below:*

$$\mathbb{E}_{p(\mathbf{x},y)}[\mathcal{L}_{ELBO}(p_\theta(\mathbf{x},y))] \leq I(y;\mathbf{z}) + I(\mathbf{x};\mathbf{u},\boldsymbol{\beta},\mathbf{z}) - [H(y) + H(\mathbf{x})] \tag{25}$$

Optimizing the ELBO of $p_\theta(\mathbf{x}, y)$ is equivalent to enhancing mutual information between label $y$ and classification latent variable $\mathbf{z}$ and between data variable $\mathbf{x}$ and all latent variables $\mathbf{u}, \boldsymbol{\beta}, \mathbf{z}$.

*Proof.* Firstly, we provide the ELBO of $\log p_\theta(\mathbf{x}, y)$ as below:

$$\begin{aligned}
\log p_\theta(\mathbf{x}, y) &= \log \int \int \int p_\theta(\mathbf{x}, \mathbf{u}, \boldsymbol{\beta}, \mathbf{z}, y) d\mathbf{z} d\boldsymbol{\beta} d\mathbf{u} \\
&= \log \int \int \int \frac{p_\theta(\mathbf{x}, \mathbf{u}, \boldsymbol{\beta}, \mathbf{z}, y) q_\phi(\mathbf{u}, \boldsymbol{\beta}, \mathbf{z}|\mathbf{x})}{q_\phi(\mathbf{u}, \boldsymbol{\beta}, \mathbf{z}|\mathbf{x})} d\mathbf{z} d\boldsymbol{\beta} d\mathbf{u} \\
&= \log \mathbb{E}_q \frac{p_\theta(\mathbf{x}, \mathbf{u}, \boldsymbol{\beta}, \mathbf{z}, y)}{q_\phi(\mathbf{u}, \boldsymbol{\beta}, \mathbf{z}|\mathbf{x})} \\
&\geq \mathbb{E}_q \log \frac{p_\theta(\mathbf{x}, \mathbf{u}, \boldsymbol{\beta}, \mathbf{z}, y)}{q_\phi(\mathbf{u}, \boldsymbol{\beta}, \mathbf{z}|\mathbf{x})} \\
&= \mathbb{E}_q[\log \frac{p_\theta(y|\mathbf{x}, \mathbf{u}, \boldsymbol{\beta}, \mathbf{z}) p_\theta(\mathbf{x}|\mathbf{u}, \boldsymbol{\beta}, \mathbf{z}) p_\theta(\mathbf{u}, \boldsymbol{\beta}, \mathbf{z})}{q_\phi(\mathbf{u}, \boldsymbol{\beta}, \mathbf{z}|\mathbf{x})}] \\
&= \mathbb{E}_q[\log p_\theta(y|\mathbf{z})] + \mathbb{E}_q[\log p_\theta(\mathbf{x}|\mathbf{u}, \boldsymbol{\beta}, \mathbf{z})] - KL[q_\phi(\mathbf{u}, \boldsymbol{\beta}, \mathbf{z}|\mathbf{x})||p_\theta(\mathbf{u}, \boldsymbol{\beta}, \mathbf{z})]
\end{aligned}$$

Then we have:

$$\mathcal{L}_{ELBO}(p_\theta(\mathbf{x}, y)) = \mathbb{E}_q \log[p_\theta(y|\mathbf{z})] + \mathbb{E}_q[\log p_\theta(\mathbf{x}|\mathbf{u}, \boldsymbol{\beta}, \mathbf{z})] - KL[q_\phi(\mathbf{u}, \boldsymbol{\beta}, \mathbf{z}|\mathbf{x})||p_\theta(\mathbf{u}, \boldsymbol{\beta}, \mathbf{z})] \tag{26}$$

To help our analysis, we introduce a helper joint distribution of variables $\mathbf{x}, y, \mathbf{z}$, which is $r(\mathbf{x}, y, \mathbf{z}) = p(\mathbf{x}, y) q_\phi(\mathbf{z}|\mathbf{x})$. This distribution has the following properties,

(1) $r(\mathbf{x}, y) = p(\mathbf{x}, y)$, $r(\mathbf{x}) = p(\mathbf{x})$, $r(y) = p(y)$.
(2) $r(\mathbf{z}|y, \mathbf{x}) = q_\phi(\mathbf{z}|\mathbf{x}) = r(\mathbf{z}|\mathbf{x})$.
(3) $r(y|\mathbf{z}, \mathbf{x}) = p(y|\mathbf{x}) = r(y|\mathbf{x})$.
(4) $I_r(y; \mathbf{z}|\mathbf{x}) = 0$, i.e. $y \perp\!\!\!\perp \mathbf{z}|\mathbf{x}$ under distribution $r$. (implied by 2, 3).

**The First Term of the ELBO**. Its upper bound is provided as follows:

$$\begin{aligned}
\mathbb{E}_{p(\mathbf{x},y)} \mathbb{E}_{q_\phi(\mathbf{z}|\mathbf{x})}[\log p_\theta(y|\mathbf{z})] &= \mathbb{E}_{r(\mathbf{x},y,\mathbf{z})}[\log p_\theta(y|\mathbf{z})] \\
&= \mathbb{E}_{r(y,\mathbf{z})}[\log p_\theta(y|\mathbf{z})] \\
&\leq \mathbb{E}_{r(y,\mathbf{z})}[\log r(y|\mathbf{z})] \\
&= \mathbb{E}_{r(y,\mathbf{z})}[\log \frac{r(y|\mathbf{z})}{p(y)} p(y)] \\
&= \mathbb{E}_{r(y,\mathbf{z})}[\log \frac{r(y|\mathbf{z})}{p(y)}] + \mathbb{E}_{p(y)}[\log p(y)] \\
&= I_r(y; \mathbf{z}) - H(y)
\end{aligned}$$

when $\log p_\theta(y|\mathbf{z}) = r(y|\mathbf{z})$, we have $\mathbb{E}_{p(\mathbf{x},y)} \mathbb{E}_{q_\phi(\mathbf{z}|\mathbf{x})}[p_\theta(y|\mathbf{z})] = I_r(y; \mathbf{z}) + H(y)$. So, $\max_{p_\theta} \mathbb{E}_{p(\mathbf{x},y)} \mathbb{E}_{q_\phi(\mathbf{z}|\mathbf{x})} \log p_\theta(y|\mathbf{z}) = I_r(y; \mathbf{z}) - H(y)$.

Then we have:

$$\mathbb{E}_{p(\mathbf{x},y)} \mathbb{E}_{q_\phi(\mathbf{z}|\mathbf{x})} \log p_\theta(y|\mathbf{z}) \leq I_r(y; \mathbf{z}) - H(y) \tag{27}$$

Furthermore, we prove another upper bound of the first term:

$$\mathbb{E}_{p(\mathbf{x},y)}\mathbb{E}_{q_\phi(\mathbf{z}|\mathbf{x})}[\log p_\theta(y|\mathbf{z})] \leq \mathbb{E}_{p(\mathbf{x},y)}\big[\log[\mathbb{E}_{q_\phi(\mathbf{z}|\mathbf{x})}p_\theta(y|\mathbf{z})]\big]$$

$$\leq \mathbb{E}_{p(\mathbf{x},y)}[\log p(y|\mathbf{x})]$$

$$= \mathbb{E}_{p(\mathbf{x},y)}[\log \frac{p(y|\mathbf{x})}{p(y)}p(y)]$$

$$= I_p(y;\mathbf{x}) - H(y)$$

The first inequality takes the equal sign when $p_\theta(y|\mathbf{z})$ is a constant w.r.t. $z$ given $x$, i.e., $p_\theta(y|\mathbf{z},\mathbf{x}) = p_\theta(y|\mathbf{x})$. The second equality takes the equal sign, when $\mathbb{E}_{q_\phi(\mathbf{z}|\mathbf{x})}p_\theta(y|\mathbf{z}) = p(y|\mathbf{x})$. We set $q_\phi(\mathbf{z}|\mathbf{x})$ and $p_\theta(y|\mathbf{z})$ as $q_\phi^*(\mathbf{z}|\mathbf{x})$ and $p_\theta^*(y|\mathbf{z})$ when the optimality of $q_\phi(\mathbf{z}|\mathbf{x})$ and $p_\theta(y|\mathbf{z})$ is achieved, i.e., $\log \mathbb{E}_{q_\phi(\mathbf{z}|\mathbf{x})}p_\theta(y|\mathbf{z}) = \log p(y|\mathbf{x}) = \mathbb{E}_{q_\phi(\mathbf{z}|\mathbf{x})}\log p_\theta(y|\mathbf{z})$. Then we have:

$$\mathbb{E}_{p(\mathbf{x},y)}\log \mathbb{E}_{q_\phi^*(\mathbf{z}|\mathbf{x})}[p_\theta^*(y|\mathbf{z})] = \mathbb{E}_{p(\mathbf{x},y)}\mathbb{E}_{q_\phi^*(\mathbf{z}|\mathbf{x})}[\log p_\theta^*(y|\mathbf{z})]$$

$$= \mathbb{E}_{p(x)}\mathbb{E}_{q_\phi^*(\mathbf{z}|\mathbf{x})}\mathbb{E}_{p_\theta^*(y|\mathbf{z})}\mathbb{E}_{q_\phi^*(\mathbf{z}|\mathbf{x})}[\log p_\theta^*(y|\mathbf{z})]$$

$$= \mathbb{E}_{p(x)}\mathbb{E}_{q_\phi^*(\mathbf{z}|\mathbf{x})}\mathbb{E}_{q_\phi^*(\mathbf{z}|\mathbf{x})}\mathbb{E}_{p_\theta^*(y|\mathbf{z})}[\log p_\theta^*(y|\mathbf{z})]$$

$$= I_*(y;\mathbf{z}) - H(y)$$

$$= I_p(y;\mathbf{x}) - H(y)$$

Thus, we prove the relationship among the two forms of upper bound for the first term and the mutual information $I_*(y;\mathbf{z})$ w.r.t. the optimal $q_\phi(\mathbf{z}|\mathbf{x})$ and $p_\theta(y|\mathbf{z})$.

$$\max_{p_\theta} \mathbb{E}_{p(\mathbf{x},y)}\mathbb{E}_{q(\mathbf{z}|\mathbf{x})}[\log p_\theta(y|\mathbf{z})] = I_r(y;\mathbf{z}) - H(y)$$

$$\leq \max_{q,p_\theta} \mathbb{E}_{p(\mathbf{x},y)}\mathbb{E}_{q(\mathbf{z}|\mathbf{x})}[\log p_\theta(y|\mathbf{z})] = I_*(y;\mathbf{z}) - H(y) = I_p(y;\mathbf{x}) - H(y)$$

Moreover, we prove the upper bound for $I_*(y;\mathbf{z})$ and $I_p(y;\mathbf{x})$:

$$I_p(y;\mathbf{x}) \leq \mathbb{E}_{q(\mathbf{x},y)}[p(y|\mathbf{x})] + H(y) \leq 0 + H(y)$$

Finally, we have $\mathbb{E}_{p(\mathbf{x},y)}\mathbb{E}_{q_\phi(\mathbf{z}|\mathbf{x})}[\log p_\theta(y|\mathbf{z})] \leq I_r(y;\mathbf{z}) \leq I_*(y;\mathbf{z}) = I_p(y;\mathbf{x}) \leq H(y)$.

**The Second Term of the ELBO**. We introduce another helper joint distribution of variables $\mathbf{x},\mathbf{u},\boldsymbol{\beta},\mathbf{z}$, which is $s(\mathbf{x},\mathbf{u},\boldsymbol{\beta},\mathbf{z}) = p(\mathbf{x})q_\phi(\mathbf{u},\boldsymbol{\beta},\mathbf{z}|\mathbf{x})$. Its upper bound is proved as follows:

$$\mathbb{E}_{p(\mathbf{x},y)}\mathbb{E}_q[\log p_\theta(\mathbf{x}|\mathbf{u},\boldsymbol{\beta},\mathbf{z})] = \mathbb{E}_{p(\mathbf{x})}\mathbb{E}_q[\log p_\theta(\mathbf{x}|\mathbf{u},\boldsymbol{\beta},\mathbf{z})]$$

$$= \mathbb{E}_{p(\mathbf{x})}\mathbb{E}_q[\log p_\theta(\mathbf{x}|\mathbf{u},\boldsymbol{\beta},\mathbf{z})]$$

$$= \mathbb{E}_{p(\mathbf{x})}\mathbb{E}_q[\log \frac{q_\phi(\mathbf{x}|\mathbf{u},\boldsymbol{\beta},\mathbf{z})}{p(\mathbf{x})}\frac{p(\mathbf{x})p_\theta(\mathbf{x}|\mathbf{u},\boldsymbol{\beta},\mathbf{z})}{q_\phi(\mathbf{x}|\mathbf{u},\boldsymbol{\beta},\mathbf{z})}]$$

$$= \mathbb{E}_{p(\mathbf{x})}\mathbb{E}_q[\log \frac{q_\phi(\mathbf{x}|\mathbf{u},\boldsymbol{\beta},\mathbf{z})}{p(\mathbf{x})}] + \mathbb{E}_{p(\mathbf{x})}\mathbb{E}_q[\log p(\mathbf{x})] + \mathbb{E}_{p(\mathbf{x})}\mathbb{E}_q[\log \frac{p_\theta(\mathbf{x}|\mathbf{u},\boldsymbol{\beta},\mathbf{z})}{q_\phi(\mathbf{x}|\mathbf{u},\boldsymbol{\beta},\mathbf{z})}]$$

$$= I_s(\mathbf{x};\mathbf{u},\boldsymbol{\beta},\mathbf{z}) - H(\mathbf{x}) + \mathbb{E}_{q_\phi(\mathbf{x},\mathbf{u},\boldsymbol{\beta},\mathbf{z})}[\log \frac{p_\theta(\mathbf{x}|\mathbf{u},\boldsymbol{\beta},\mathbf{z})}{q_\phi(\mathbf{x}|\mathbf{u},\boldsymbol{\beta},\mathbf{z})}]$$

$$= I_s(\mathbf{x};\mathbf{u},\boldsymbol{\beta},\mathbf{z}) - H(\mathbf{x}) + \mathbb{E}_{q_\phi(\mathbf{u},\boldsymbol{\beta},\mathbf{z})}\mathbb{E}_{q_\phi(\mathbf{x}|\mathbf{u},\boldsymbol{\beta},\mathbf{z})}[\log \frac{p_\theta(\mathbf{x}|\mathbf{u},\boldsymbol{\beta},\mathbf{z})}{q_\phi(\mathbf{x}|\mathbf{u},\boldsymbol{\beta},\mathbf{z})}]$$

$$= I_s(\mathbf{x};\mathbf{u},\boldsymbol{\beta},\mathbf{z}) - H(\mathbf{x}) - \mathbb{E}_{q_\phi(\mathbf{u},\boldsymbol{\beta},\mathbf{z})}\big[KL[q_\phi(\mathbf{x}|\mathbf{u},\boldsymbol{\beta},\mathbf{z})||p_\theta(\mathbf{x}|\mathbf{u},\boldsymbol{\beta},\mathbf{z})]\big]$$

$$\leq I_s(\mathbf{x};\mathbf{u},\boldsymbol{\beta},\mathbf{z}) - H(\mathbf{x}) - 0$$

where $I_s(\mathbf{x};\mathbf{u},\boldsymbol{\beta},\mathbf{z})$ is w.r.t. $s(\mathbf{x},\mathbf{u},\boldsymbol{\beta},\mathbf{z})$, i.e., $p(\mathbf{x})q_\phi(\mathbf{u},\boldsymbol{\beta},\mathbf{z}|\mathbf{x})$.

Then we have:

$$\mathbb{E}_{p(\mathbf{x},y)}\mathbb{E}_q[\log p_\theta(\mathbf{x}|\mathbf{u},\boldsymbol{\beta},\mathbf{z})] \leq I_s(\mathbf{x};\mathbf{u},\boldsymbol{\beta},\mathbf{z}) - H(\mathbf{x}) \tag{28}$$

Applying Eqn. 27 and Eqn. 28 to Eqn. 26, we have:

$$\mathbb{E}_{p(\mathbf{x},y)}\mathcal{L}_{ELBO}(p_\theta(\mathbf{x},y))$$

$$= \mathbb{E}_{p(\mathbf{x},y)}\mathbb{E}_q[\log p_\theta(y|\mathbf{z})] + \mathbb{E}_{p(\mathbf{x},y)}\mathbb{E}_q[\log p_\theta(\mathbf{x}|\mathbf{u},\boldsymbol{\beta},\mathbf{z})] - \mathbb{E}_{q_\phi(\mathbf{u},\boldsymbol{\beta},\mathbf{z})}\big[KL[q_\phi(\mathbf{x}|\mathbf{u},\boldsymbol{\beta},\mathbf{z})||p_\theta(\mathbf{x}|\mathbf{u},\boldsymbol{\beta},\mathbf{z})]\big]$$

$$\leq I_r(y;\mathbf{z}) + I_s(\mathbf{x};\mathbf{u},\boldsymbol{\beta},\mathbf{z}) - [H(y) + H(\mathbf{x})]$$

For clarity, we use $I(y; \mathbf{z})$ and $I(\mathbf{x}; \mathbf{u}, \boldsymbol{\beta}, \mathbf{z})$ in place of $I_r(y; \mathbf{z})$ and $I_s(\mathbf{x}; \mathbf{u}, \boldsymbol{\beta}, \mathbf{z})$ ,repectively, later in the Appendix and the main paper. $\square$

**Lemma B.2** (**Information Decomposition of the Adversarial Loss**). *The global maximum of adversarial loss w.r.t. discriminator $D$ is decomposed as below:*

$$\max_D \mathbb{E}_{p(k,\mathbf{x})}\mathbb{E}_{q_\phi(\mathbf{z}|\mathbf{x})}[\log D(k|\mathbf{z})] = I(\mathbf{z}; \boldsymbol{\beta}) + I(\mathbf{z}, k|\boldsymbol{\beta}) - H(k) \tag{29}$$

*and the global minimum of $\max_D \mathbb{E}_{p(k,\mathbf{x})}\mathbb{E}_{q_\phi(\mathbf{z}|\mathbf{x})}[\log D(k|\mathbf{z})]$ is achieved if and only if $I(\mathbf{z}; \boldsymbol{\beta}) = I(\mathbf{z}, k|\boldsymbol{\beta}) = 0$.*

The optimization of adversarial loss is to reduce the mutual information between data encoding $\mathbf{z}$ and domain identity $k$ while the mutual information is reduced between data encoding $\mathbf{z}$ and domain identity $\boldsymbol{\beta}$.

*Proof.* Define

$$s(k, \mathbf{x}, \boldsymbol{\beta}, \mathbf{z}) = p(k, \mathbf{x})q_\phi(\mathbf{z}|\mathbf{x})q_\phi(\boldsymbol{\beta}|k)$$
$$s(k, \boldsymbol{\beta}, \mathbf{z}) = q_\phi(\boldsymbol{\beta}, \mathbf{z}|k)p(k) = q_\phi(\mathbf{z}|\boldsymbol{\beta}, k)q_\phi(\boldsymbol{\beta}|k)p(k)$$
$$s(k, \mathbf{z}) = p(k)q_\phi(\mathbf{z}|k) = s(k|\mathbf{z})q_\phi(\mathbf{z}) = p(k)\mathbb{E}_{p(\mathbf{x}|k)}\mathbb{E}_{q_\phi(\mathbf{z}|\mathbf{x})}$$

Then we have:

$$-\mathbb{E}_{p(k,\mathbf{x})}\mathbb{E}_{q_\phi(\mathbf{z}|\mathbf{x})}[\log D(k|\mathbf{z})] = -\mathbb{E}_{s(\mathbf{z},k)}[\log D(k|\mathbf{z})] \leq -\mathbb{E}_{s(\mathbf{z},k)}[\log s(k|\mathbf{z})]$$

When the equality holds, the discriminator becomes optimal, i.e., $D(k|\mathbf{z}) = s(k|\mathbf{z})$. Due to $p(\boldsymbol{\beta}|k)$ is a function $f(k)$ mapping from $k$ to $\boldsymbol{\beta}$. Therefore, we have:

$$p(\mathbf{z}|\boldsymbol{\beta}, k) = p(\mathbf{z}|f(k), k) = p(\mathbf{z}|k)$$

Thus, $s(k, \boldsymbol{\beta}, \mathbf{z}) = q_\phi(\mathbf{z}|k)q_\phi(\boldsymbol{\beta}|k)p(k)$. The three random variables meet the Markov chain $\boldsymbol{\beta} \leftarrow k \rightarrow \mathbf{z}$. By chain rule for mutual information, we have:

$$I(\mathbf{z}; \boldsymbol{\beta}) + I(\mathbf{z}, k|\boldsymbol{\beta}) = I(\mathbf{z}; \boldsymbol{\beta}, k) = I(\mathbf{z}; k) + I_q(\mathbf{z}, \boldsymbol{\beta}|k)$$

where $I_q(\mathbf{z}, \boldsymbol{\beta}|k) = 0$. So,

$$I(\mathbf{z}; k) = I(\mathbf{z}; \boldsymbol{\beta}) + I(\mathbf{z}, k|\boldsymbol{\beta})$$

We also have:

$$\mathbb{E}_{s(\mathbf{z},k)}[\log s(k|\mathbf{z})] = \mathbb{E}_{s(\mathbf{z},k)}[\log \frac{s(k|\mathbf{z})}{q_\phi(k)}q_\phi(k)]$$
$$= \mathbb{E}_{s(\mathbf{z},k)}[\log \frac{s(k|\mathbf{z})}{q_\phi(\mathbf{z})}] + \mathbb{E}_{s(\mathbf{z},k)}[\log q_\phi(k)]$$
$$= I(\mathbf{z}; k) - H(k)$$

$$\max_D \mathbb{E}_{p(k,\mathbf{x})}\mathbb{E}_{q_\phi(\mathbf{z}|\mathbf{x})}[\log D(k|\mathbf{z})] = I(\mathbf{z}; k) - H(k) = I(\mathbf{z}; \boldsymbol{\beta}) + I(\mathbf{z}, k|\boldsymbol{\beta}) - H(k)$$

Therefore, $\min_\phi \max_D \mathbb{E}_{p(k,\mathbf{x})}\mathbb{E}_{q_\phi(\mathbf{z}|\mathbf{x})}[\log D(k|\mathbf{z})] = 0 - H(k)$ while $I(\mathbf{z}; k) = 0$ and thus $I(\mathbf{z}; \boldsymbol{\beta}) = I(\mathbf{z}, k|\boldsymbol{\beta}) = 0$. $\square$

**Theorem B.1** (**Objective Function as a Lower Bound**). *The objective function involves both the ELBO of $p_\theta(\mathbf{x}, y)$ and adversarial loss $\mathbb{E}_{p(k,\mathbf{x})}\mathbb{E}_{q_\phi(\mathbf{z}|\mathbf{x})}[\log D(k|\mathbf{z})]$, and it is the lower bound for a combination mutual information and entropy terms:*

$$\mathbb{E}_{p(\mathbf{x},y)}[\mathcal{L}_{ELBO}(\mathbf{x}, y)] - \max_D \mathbb{E}_{p(k,\mathbf{x})}\mathbb{E}_{q_\phi(\mathbf{z}|\mathbf{x})}[\log D(k|\mathbf{z})] \tag{30}$$

$$\leq I(y; \mathbf{z}) + I(\mathbf{x}; \mathbf{u}, \boldsymbol{\beta}, \mathbf{z}) - I(\mathbf{z}; \boldsymbol{\beta}) - I(\mathbf{z}, k|\boldsymbol{\beta}) - [H(y) + H(\mathbf{x}) - H(k)]. \tag{31}$$

*Proof.* Theorem B.1 is proved by applying Eqn. 25 mine Eqn. 29. $\square$

**Theorem B.2** (**Global Optimum of VDI**). *In VDI, the global optimum (Eqn. 20) is achieved, if and only if (1) $I(\mathbf{z}; \boldsymbol{\beta}) = I(\mathbf{z}, k; \boldsymbol{\beta}) = 0$, (2) $I(\mathbf{x}; \mathbf{u}, \boldsymbol{\beta}, \mathbf{z})$ is maximized, and (3) $I(y; \mathbf{z})$ is maximized.*

*Proof.* The condition (1) is gotten from Lemma B.2. The last two conditions (2) and (3) are gotten from Lemma B.1. □

As Theorem B.2 states, the global optimum of Eqn. 20 is guaranteed to satisfy all three conditions in Definition 3.1; therefore training VDI using the minimax game objective Eqn. 20 is equivalent to inferring the optimal domain indices.

## C   VISUALIZATION OF INFERRED DOMAIN INDICES FOR CIRCLE

Fig. 7 shows the inferred domain indices for Circle.

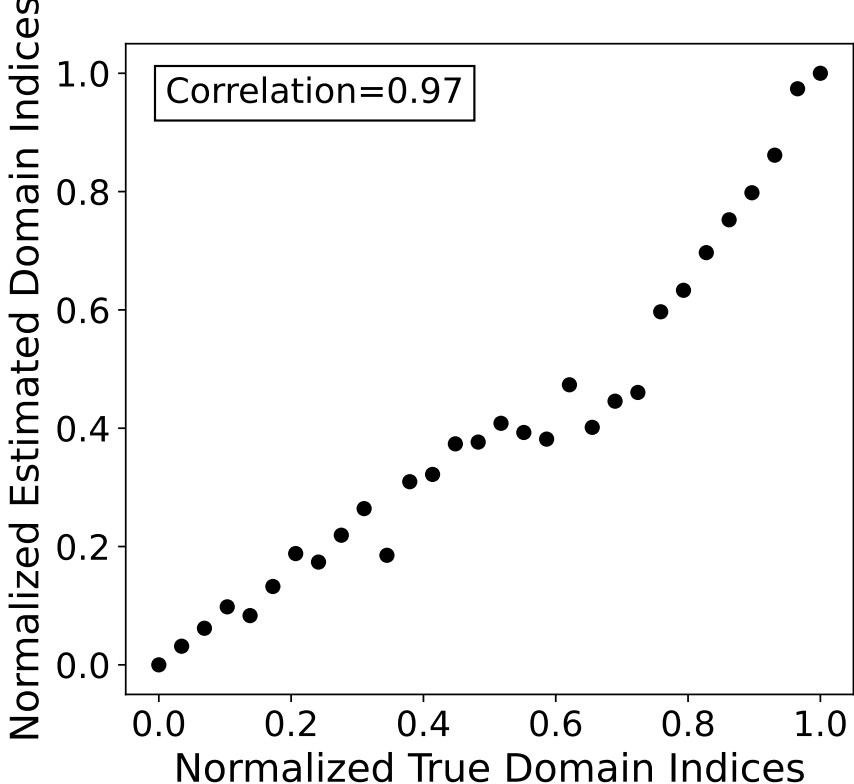

Figure 7:   Inferred domain indices (reduced to 1 dimension by PCA) with true domain indices for dataset *Circle*. VDI's inferred indices have a correlation of 0.97 with true indices, even though *VDI does not have access to true indices during training*.

## D   VISUALIZATION OF LOCAL DOMAIN INDICES FOR DG-15

Fig. 8 shows the local domain indices for DG-15. This inspired us to derive global domain index by Earth Moving distance.

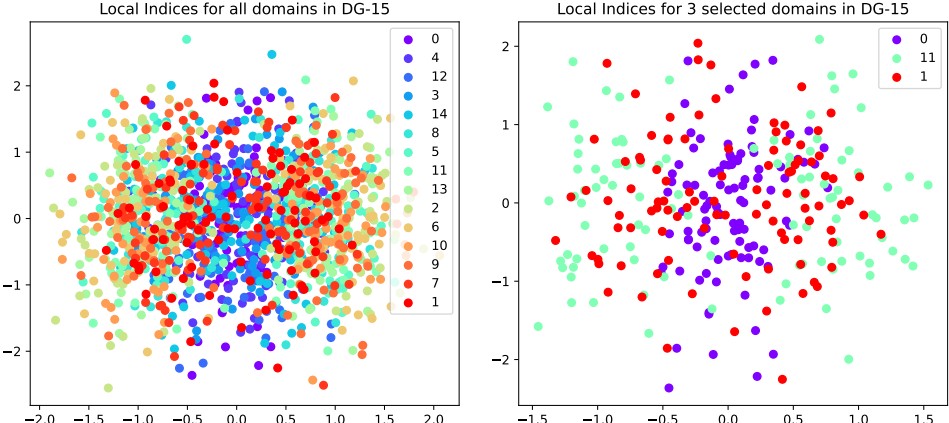

Figure 8: Local Domain Indices for *DG-15*. **Left:** Local domain indices for every domain. **Right:** Local domain indices for 3 selected domain. We select each domain by their location in domain graph (see Fig. 9(a))

# E ARCHITECTURE AND IMPLEMENTATION DETAILS

## E.1 ARCHITECTURE

As is shown in Fig. 2, we use neural networks to estimate the density function of each distribution. For *Circle*, *DG-15*, *DG-60* and *TPT-48*, we use multi-layer perceptrons for estimating $q(\mathbf{u}|\mathbf{x})$, while for *CompCars*, we use ResNet-18 (He et al., 2015) for $q(\mathbf{u}|\mathbf{x})$. All the other neural networks are multi-layer perceptrons. To ensure the robustness of training, we fix the variance of some distributions and only use a neural network to estimate the mean of these distributions. All the input data are normalized with their mean and variance.

## E.2 HYPERPARAMETERS

For experiments on all 4 datasets, we set the dimension of global domain indices to 2. For *Circle*, *DG-15*, *DG-60*, the dimension of local domain indices is 4, while for *TPT-48* and *CompCars*, the dimension of local domain indices is 8. Our model is trained with 20 to 70 warmup steps, learning rates ranging from $1 \times 10^{-5}$ to $1 \times 10^{-4}$, and $\lambda_d$ ranging from 0.1 to 1.

## E.3 ADDITIONAL LOSS ON LOCAL DOMAIN INDICES

During the inference of local domain indices $\mathbf{u}$, we incorporate a modified contrastive loss from Chen et al. (2020) to improve the coherence of $\mathbf{u}$. The loss aims to maximize the agreement between local domain indices of data points in the same domain. Specifically, we sampled a minibatch of size $b$ from each of the $N$ domains, resulting in a large batch of size $bN$. For each $\mathbf{x}_{k,i}$, the $i$-th sample in the minibatch of domain $k$, we pair it with $\mathbf{x}_{k,(i+1) \bmod b}$, the neighbouring data point within the same domain. We denote $j = (i + 1) \bmod b$, and refer to the sampled pair as $(\mathbf{x}_{k,i}, \mathbf{x}_{k,j})$. We then sample corresponding $\mathbf{u}_{k,i}, \mathbf{u}_{k,j}$ from $q_\phi(\mathbf{u}|\mathbf{x})$ (Eqn. 6). With a multi-layer perceptron, we map $\mathbf{u}_{k,i}, \mathbf{u}_{k,j}$ to $\mathbf{h}_{k,i}, \mathbf{h}_{k,j}$, which are subsequently used to calculate the contrastive loss as follows:

$$\ell_{k,i}^{\text{Con}} = -\log \frac{\exp(\text{sim}(\mathbf{h}_{k,i}, \mathbf{h}_{k,j})/\tau)}{\sum_{m=1}^{N} \sum_{n=1}^{b} \mathbb{1}_{k \neq m} \exp(\text{sim}(\mathbf{h}_{k,i}, \mathbf{h}_{m,n})/\tau)},$$

where $\text{sim}(\cdot, \cdot)$ is the similarity function between 2 vectors, and $\tau$ is the temperature. In practise, we use the cosine similarity and set $\tau = 1$.

# F DATASET SUMMARY

Table 4 summarizes the statistics for all the datasets used in our experiments.

Table 4: Summary of statistics and settings in different datasets.

| Dataset | Number of Samples | Input Dim | Toy/Real | Adaptation Task |
|---------|-------------------|-----------|----------|-----------------|
| *Circle* | 3,000 | 2 | Toy | 2-Way Classification |
| *DG-15* | 1,500 | 2 | Toy | 2-Way Classification |
| *DG-60* | 6,000 | 2 | Toy | 2-Way Classification |
| *TPT-48* | 6,912 | 6 | Real | Regression |
| *CompCars* | 18,735 | $224 \times 224$ | Real | 4-Way Classification |

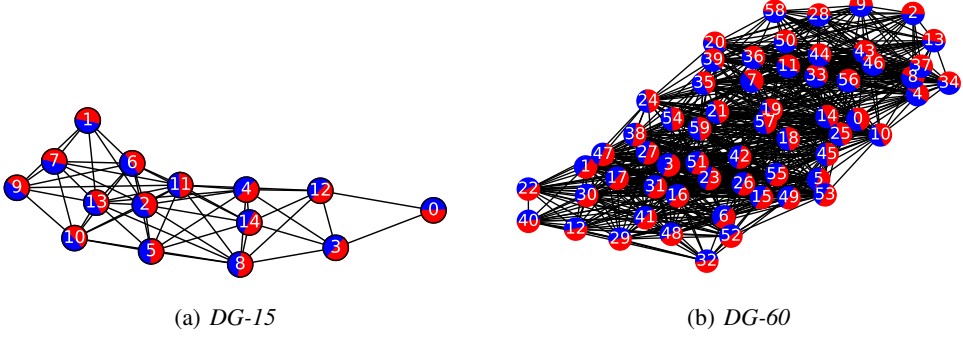

(a) *DG-15*        (b) *DG-60*

Figure 9: Visualization of the *DG-15* (a) and *DG-60* (b) datasets. We use 'red' and 'blue' to indicate positive and negative data points inside a domain. The boundaries between 'red' half circles and 'blue' half circles show the direction of ground-truth decision boundaries in the datasets.

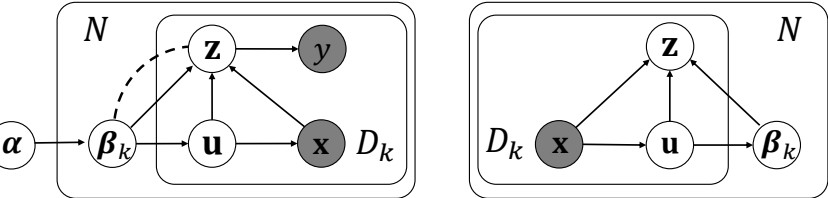

Figure 10: (Copied Fig. 1 here for easier reference.) **Left:** Probabilistic graphical model for VDI's generative model. We introduce a new edge type, "---", to denote independence. $\boldsymbol{\beta}_k$ --- $\mathbf{z}$ enforces independence between $\mathbf{z}$ and $\boldsymbol{\beta}_k$, i.e, $p(\mathbf{z}|\boldsymbol{\beta}_k) = p(\mathbf{z})$; note that this does *not* contradict $\boldsymbol{\beta}_k \rightarrow \mathbf{z}$ in our case. **Right:** Probabilistic graphical model for the VDI's inference model.

## G   DG-15 AND DG-60

For completeness, we show *DG-15* and *DG-60* in Fig. 9. We use 'red' and 'blue' to indicate positive and negative data points inside a domain. The boundaries between 'red' half circles and 'blue' half circles show the direction of ground-truth decision boundaries in the datasets.

*DG-15* is a synthetic dataset with 15 domains for binary classification. As shown in Fig. 3(c), these domains form a domain graph (DG) of 15 nodes, with adjacent domains having similar decision boundaries. Each domain contains 100 data points. We use 6 connected domains as the source domains and use others as target domains. Similarly, *DG-60* is another synthetic dataset with 60 domains, each of which contains 100 data points. We use 6 connected domains as source domains and the remaining 54 domains as target domains.

## H   ADAPTING MDD AND SENTRY FOR REGRESSION TASKS

For **MDD**, we simply replace its cross-entropy loss with an $L_2$ loss. For **SENTRY**, since it requires confidence scores during training, we include another classification network that predicts whether the average temperature of next 6 months will go up or down compared to the previous 6 months, thereby providing confidence scores.

## I   NEW EDGE TYPE FOR PROBABILISTIC GRAPHICAL MODELS

In Fig. 10, we introduce a new edge type, "---", to denote independence. $\boldsymbol{\beta}_k$ --- $\mathbf{z}$ enforces independence between $\mathbf{z}$ and $\boldsymbol{\beta}_k$, i.e, $p(\mathbf{z}|\boldsymbol{\beta}_k) = p(\mathbf{z})$; note that this does *not* contradict $\boldsymbol{\beta}_k \to \mathbf{z}$, as long as there are multiple paths between $\boldsymbol{\beta}_k$ and $\mathbf{z}$. In this case, we have two paths between $\boldsymbol{\beta}_k$ and $\mathbf{z}$, $\boldsymbol{\beta}_k \to \mathbf{z}$ and $\boldsymbol{\beta}_k \to \mathbf{u} \to \mathbf{z}$.

As a simplified example, consider only a sub-graph in Fig. 10 with only three nodes $\boldsymbol{\beta}_k$, $\mathbf{u}$, and $\mathbf{z}$. Essentially our algorithm tries to learn two conditional dependencies $p(\mathbf{u}|\boldsymbol{\beta}_k)$ and $p(\mathbf{z}|\boldsymbol{\beta}_k, \mathbf{u})$. Adding a dashed edge $\boldsymbol{\beta}_k$ --- $\mathbf{z}$ ensures that $p(\mathbf{z}|\boldsymbol{\beta}_k) = p(\mathbf{z})$, which is equivalent to

$$p(\mathbf{z}|\boldsymbol{\beta}_k) = p(\mathbf{z}|\boldsymbol{\beta}_k'), \ \ \forall \boldsymbol{\beta}_k, \boldsymbol{\beta}_k',$$

which is equivalent to

$$\int p(\mathbf{u}|\boldsymbol{\beta}_k)p(\mathbf{z}|\boldsymbol{\beta}_k, \mathbf{u})d\mathbf{u} = \int p(\mathbf{u}|\boldsymbol{\beta}_k')p(\mathbf{z}|\boldsymbol{\beta}_k', \mathbf{u})d\mathbf{u}, \ \ \forall \boldsymbol{\beta}_k, \boldsymbol{\beta}_k'. \tag{32}$$

It is easy to see that Eqn. 32 only introduces an additional constraint, but does not contradict the learning of conditional dependencies $p(\mathbf{u}|\boldsymbol{\beta}_k)$ and $p(\mathbf{z}|\boldsymbol{\beta}_k, \mathbf{u})$.

Combining the ELBO below

$$\mathcal{L}_{ELBO}(\mathbf{x}, y) = \mathbb{E}_{q_\phi(\mathbf{u},\boldsymbol{\beta},\mathbf{z}|\mathbf{x})}[p_\theta(\mathbf{u}, \mathbf{x}, \mathbf{z}, y, \boldsymbol{\beta}|\boldsymbol{\alpha})] - \mathbb{E}_{q_\phi(\mathbf{u},\boldsymbol{\beta},\mathbf{z}|\mathbf{x})}[q_\phi(\mathbf{u}, \boldsymbol{\beta}, \mathbf{z}|\mathbf{x})], \tag{33}$$

with the additional constraint in Eqn. 32, we have the following objective function

$$\max_{\boldsymbol{\theta},\boldsymbol{\phi}} \ \mathbb{E}_{p(\mathbf{x},y)}\Big[\mathbb{E}_{q_\phi(\mathbf{u},\boldsymbol{\beta},\mathbf{z}|\mathbf{x})}[p_\theta(\mathbf{u}, \mathbf{x}, \mathbf{z}, y, \boldsymbol{\beta}|\boldsymbol{\alpha})] - \mathbb{E}_{q_\phi(\mathbf{u},\boldsymbol{\beta},\mathbf{z}|\mathbf{x})}[q_\phi(\mathbf{u}, \boldsymbol{\beta}, \mathbf{z}|\mathbf{x})]\Big], \tag{34}$$

$$s.t. \int p_\theta(\mathbf{u}|\boldsymbol{\beta}_k)p_\theta(\mathbf{z}|\boldsymbol{\beta}_k, \mathbf{u})d\mathbf{u} = \int p_\theta(\mathbf{u}|\boldsymbol{\beta}_k')p_\theta(\mathbf{z}|\boldsymbol{\beta}_k', \mathbf{u})d\mathbf{u}, \ \ \forall \boldsymbol{\beta}_k, \boldsymbol{\beta}_k', \tag{35}$$

which is equivalent to Eqn. 20.

## J   DOMAIN IDENTITIES VERSUS DOMAIN INDICES

Compared to domain identities, domain indices can better capture the similarity and complex relations among different domains, thereby better guiding the adaptation process. This is also empirically verified in previous works such as Wang et al. (2020); Xu et al. (2022). As a simple example, note that domain identities are one-hot vectors, the distance between any two domain identities are identical; in contrast, domain indices are real-value vectors and therefore contain richer information.

More specifically, since the encoder takes domain indices as additional input, domain indices tend to encourage data $\mathbf{x}$ of similar domains to go through similar transformations to produce the encoding $\mathbf{z}$; consequently, the predicted decision boundaries of similar domains will also be similar.

Theoretically, the target domain's error is upper-bounded by three terms, i.e., source error, domain gap, and optimal joint error of the source and target domains (Ben-David et al., 2010). Encouraging similar transformations for similar domains can reduce the joint error term of the upper bound, thereby achieving better performance.

## K   GLOBAL VERSUS LOCAL DOMAIN INDICES

**Global versus Local Domain Indices.** Local domain indices $\mathbf{u}$ contain instance-level information, i.e., each data point has a different local domain index $\mathbf{u}$; in contrast, global domain indices contain domain-level information, i.e., all data points of the same domain share the same global domain index $\boldsymbol{\beta}$.

**Necessity of Local Domain Indices.** Essentially, local domain indices $\mathbf{u}$ serve as extremely low-dimensional summaries of representations $\mathbf{z}$; for example, in CompCars experiments the numbers of dimensions for $\mathbf{u}$ and $\mathbf{z}$ are 4 and 4096, respectively. With such low-dimensional $\mathbf{u}$, global domain indices can be efficiently inferred from $\mathbf{u}$ in the 4-dimensional space using Eq. (8-11). In contrast, if one only uses global indices without $\mathbf{u}$, one would need to compute the Earth Mover's distance

(EMD) and multi-dimensional scaling (MDS) in Eq. (8-11) in the $4096$-dimensional space; this is much more computationally expensive and dramatically slows down model training. Therefore local domain indices are necessary in VDI.

**Necessity of EMD and MDS.** It is also worth noting that EMD and MDS are also necessary when inferring global domain indices $\beta$ from local domain indices $\mathbf{u}$. The main reason is that $\mathbf{u}$'s distribution tends to be multi-modal. As an example, we plot the local indices from 3 domains in *DG-15* in Fig. 8 of Appendix D. We can see that the $\mathbf{u}$'s in Domain 11 contain two clusters (i.e. two modes), with one cluster corresponding to positive data and the other corresponding to negative data. This is also the case for both Domain 1 and 0, but the distance between the two clusters gets smaller and smaller. Therefore, directly using the mean of $u$'s as the global domain index $\beta$ does not work because all three domains will have similar means, and consequently similar $\beta$'s. Using EMD and MDS fixes this issue since EMD can naturally compute the distance between two multi-modal distributions. Our preliminary experiments also confirm their necessity; removing EMD and MDS will significantly bring down VDI's performance.

**Independence between z and the Global Index $\beta$.** In typical domain adaptation methods, an encoder first takes $\mathbf{x}$ as input to produce the representation $z$, and a predictor then takes $\mathbf{z}$ as input to predict the label. To improve $\mathbf{z}$'s generalization across different domains, it is common practice (Ganin et al., 2016; Zhao et al., 2017; Zhang et al., 2019; Tzeng et al., 2017; Wang et al., 2020; Xu et al., 2022) to enforce independence between the domain index $\beta$ and representation $\mathbf{z}$ such that domain-specific information is removed from $\mathbf{z}$. Therefore the assumption/constraint of independence between $\beta$ and $\mathbf{z}$ is natural.

**Why Independence between z and the Local Index $u$ is Not Required.** As mentioned above, we need the local indices $u$ to capture the multi-cluster structure in the data, with each cluster corresponding to local indices for data with the same label $y$. Since $\mathbf{z}$ contains label-specific information, if we enforce independence between $\mathbf{z}$ and $\mathbf{u}$, such label-specific information will be removed from $\mathbf{u}$, making it impossible for $\mathbf{u}$ to capture the label-specific multi-cluster structure in the data.

**How the Global Index $\beta$ Indicates Different Domains from the Definition.** One key property that makes sure the global index $\beta$ can distinguish different domains is the second point of Definition 3.1, i.e., information preservation of $\mathbf{x}$. Specifically, maximizing the mutual information $I(\mathbf{x}; \mathbf{u}, \beta, \mathbf{z})$ ensures that $\beta$ contains as much information on $x$ as possible under the constraint that $\beta$ and $\mathbf{z}$ is independent. Since we assume covariant shift exists (i.e., different domains have different distributions over $\mathbf{x}$), this property helps $\beta$ to distinguish (indicate) different domains. This is also empirically verified by results in Fig. 3(d), Fig. 5, and Fig. 6, where VDI successfully inferred meaningful domain indices $\beta$ from different datasets.

## L   LARGER FIGURES

In this section, we provide larger versions of figures for domain index visualization in the main paper.

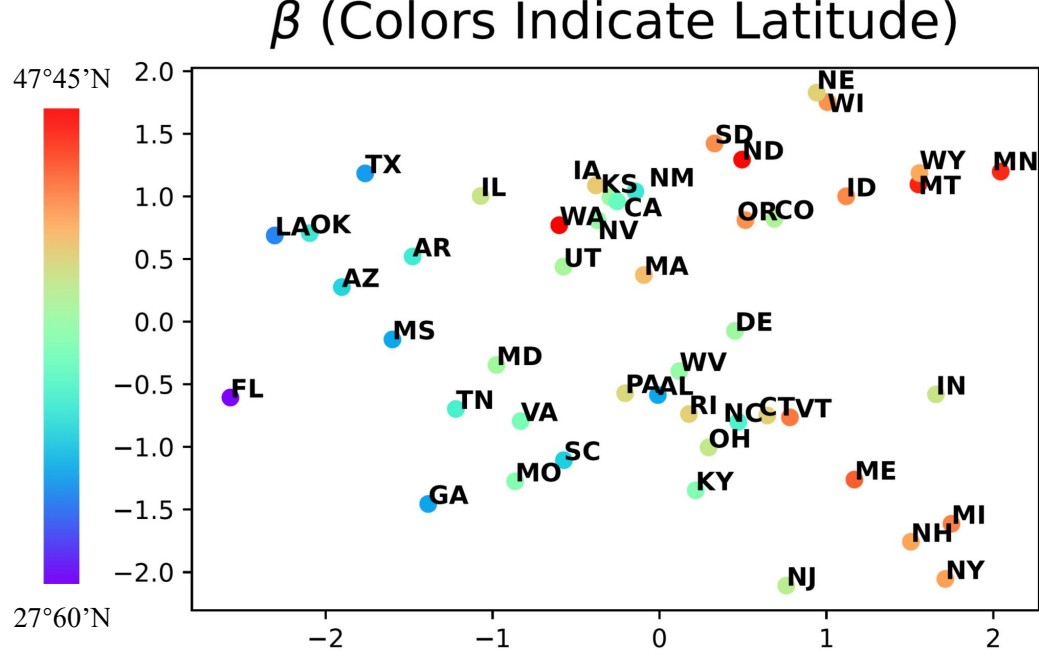

Figure 11: Inferred domain indices for 48 domains in *TPT-48*. We color inferred domain indices according to ground-truth latitude. VDI's inferred indices are correlated with true indices, even though *VDI does not have access to true indices during training*.

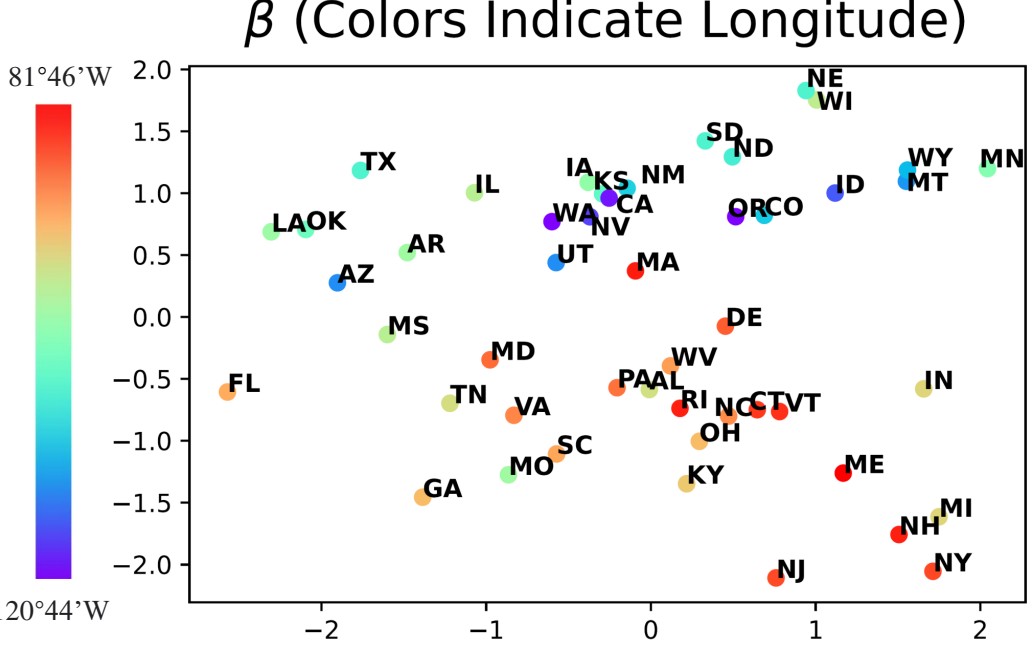

Figure 12: Inferred domain indices for 48 domains in *TPT-48*. We color inferred domain indices according to ground-truth longitude. VDI's inferred indices are correlated with true indices, even though *VDI does not have access to true indices during training*.

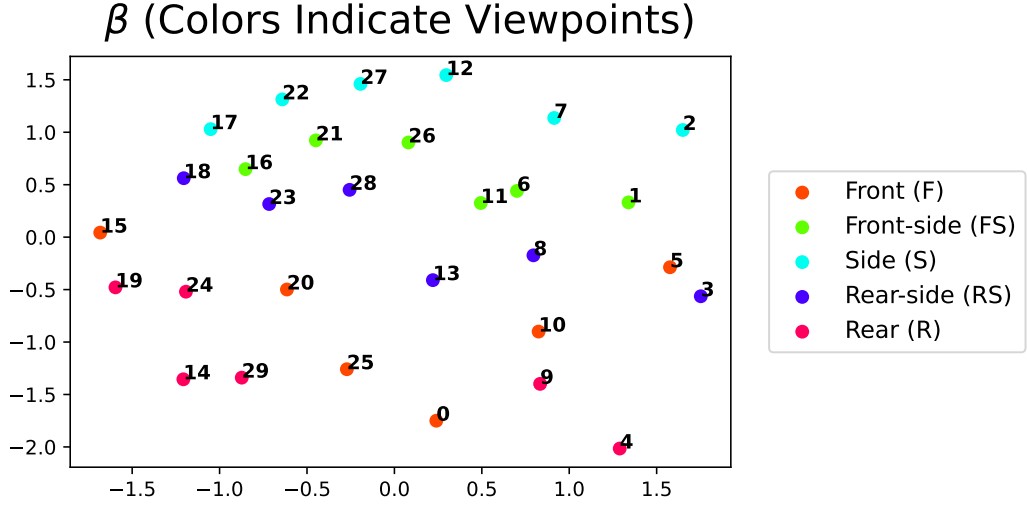

Figure 13: Inferred domain indices for 30 domains in *CompCars*. We color inferred domain indices according to ground-truth viewpoints. VDI's inferred indices are correlated with true indices, even though *VDI does not have access to true indices during training*.

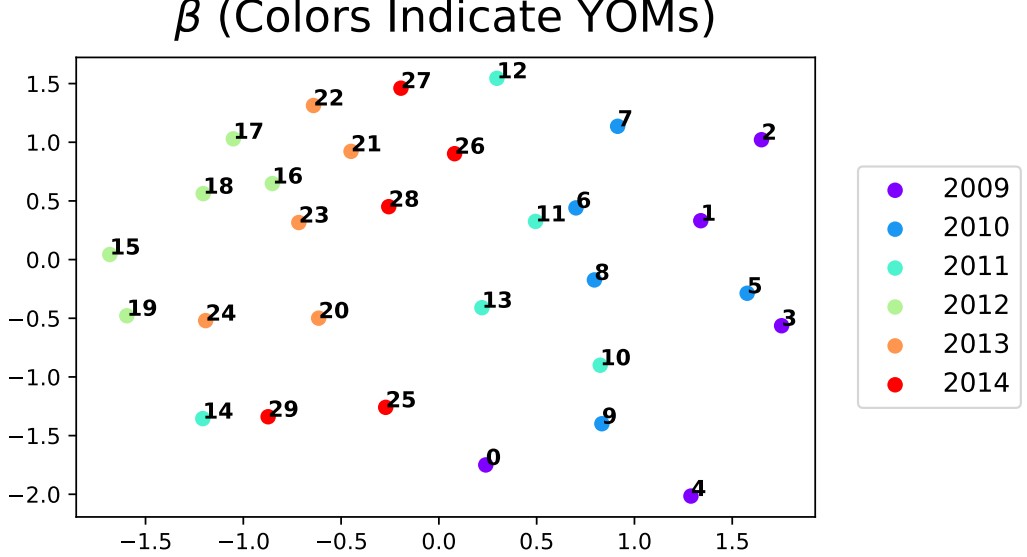

Figure 14: Inferred domain indices for 30 domains in *CompCars*. We color inferred domain indices according to ground-truth years of manufacture (YOMs). VDI's inferred indices are correlated with true indices, even though *VDI does not have access to true indices during training*.

