# OpenReview forum: "Domain-Indexing Variational Bayes: Interpretable Domain Index for Domain Adaptation"
_ICLR.cc/2023/Conference — ICLR 2023 notable top 25%_

### Official Review · Reviewer_YbH3 · 2022-10-15

**Confidence:** 4
**Correctness:** 3
**Technical Novelty And Significance:** 3
**Empirical Novelty And Significance:** 4
**Recommendation:** 8

**Clarity, Quality, Novelty And Reproducibility:**

The paper is clear and very well written. To the best of my knowledge, this is the first paper proposing to learn continuous domain indexes for DA directly from data. As mentioned before, some details are missing for reproducibility.

**Strength And Weaknesses:**

Strengths:
- The paper is well-motivated and its contributions are presented clearly.
- The proposed method is technically sound and elegant, although some relevant details seem to be missing (see 'Weaknesses' below).
- The methodology is well grounded by the presented theoretical results.
- The experiments confirm the effectiveness of the method. In particular, the visualizations of the learned global domain indexes in Figures 5 and 6 are quite interesting.

Weaknesses:
1. As is apparent from section 3.1, the paper considers the unsupervised multi-source, multi-target DA problem. However, it is not entirely clear which terms in the loss (eq. (14)) are active for unlabeled target examples. I suppose it should be all the terms of the ELBO except for eq. (5), plus the domain discriminator loss, but this should be clarified. From a theoretical perspective, I think this would correspond to the maximization of the ELBO for $p(x)$ plus domain discriminator loss.
2. In the proof of Lemma A.1, the authors claim that "since $z_1 \perp \beta$ and $z_2 \perp \beta$, then $(z_1, z_2) \perp \beta$", but this is not true in general. E.g. suppose $z_1$ and $z_2$ are Bernoulli r.v.'s and $\beta = z_1  \mathrm{xor}  z_2$. The authors should explain why the implication holds in their case or otherwise review the proof and reformulate the remainder of the theoretical analysis in case the Lemma cannot be proved.
3. Architecture and implementation details are missing from the experimental section. This should be provided as supplementary material. Releasing the source code would also be beneficial for reproducibility.


**Summary Of The Paper:**

The paper proposes a domain adaptation (DA) method that, as an additional outcome, learns continuous domain indexes directly from multi-domain data and annotated domain identities. Experiments show that the learned indexes improve the predictive performance of the model in DA. The method relies on a graphical model to describe the joint distribution of inputs $x$, targets $y$, latent domain indexes $\beta$ and $u$, and domain-independent latent codes $z$. Learning is performed by the maximization of the usual ELBO for the likelihood $p(x, y)$. In addition, domain-independent latent codes $z$ are enforced by an adversarial domain discriminator.

**Summary Of The Review:**

This well-written paper presents solid contributions to the challenging problem of unsupervised multi-source, multi-target DA. Weaknesses 1. and 3. can be solved easily. Weakness 2. might be harder to solve, but I think that, even if Lemma A.1 cannot be proved and has to be removed, this result is not crucial for the paper. Thus, I recommend acceptance on the condition that the authors will address these concerns appropriately in their rebuttal and change the manuscript accordingly.

---

> ### Author Response · Authors · 2022-11-15
> **Thank you for the constructive and encouraging comments**
>
> Thank you for your constructive and encouraging comments as well as the insightful questions. We are glad that you find our paper ``"well-motivated"``, our contributions ``"presented clearly"``/``"solid"``, our methodology ``"well grounded by the presented theoretical results"``/``"elegant"``, and our experiments ``"interesting"``. Below we address your questions in detail one by one.
>
> **Q1. As is apparent from section 3.1, the paper considers the unsupervised multi-source, multi-target DA problem. However, it is not entirely clear which terms in the loss (eq. (14)) are active for unlabeled target examples. I suppose it should be all the terms of the ELBO except for eq. (5), plus the domain discriminator loss, but this should be clarified. From a theoretical perspective, I think this would correspond to the maximization of the ELBO for p(x) plus domain discriminator loss.**
>
> Yes, you are correct. We follow the typical domain adaptation setting to use labeled data $(x, y)$ from source domains and unlabeled data $x$ from target domains during training. For source-domain data, we use the full ELBO, i.e., Eq (4-7), plus the discriminator loss; for target-domain data, we use Eq (4), (6), and (7) plus the discriminator loss. We have revised the paper accordingly as suggested to make this clear.
>
> **Q2. In the proof of Lemma A.1, the authors claim that " since z1⊥β and z2⊥β, then (z1,z2)⊥β", but this is not true in general. E.g. suppose z1 and z2 are Bernoulli r.v.'s and β=z1 xor z2. The authors should explain why the implication holds in their case or otherwise review the proof and reformulate the remainder of the theoretical analysis in case the Lemma cannot be proved.**
>
> Thanks for mentioning this and we apologize for the confusion. Indeed, for $(z_1, z_2)⊥\beta$ to hold, one needs an additional assumption $z_1 ⊥ z_2 | \beta$. We therefore revised Lemma A.1 and provided proof that does not need such an assumption.
>
> Note that (and as you correctly pointed out) the revision on Lemma A.1 does not affect any conclusions or claims in the main paper. Lemma A.1 is only meant for a more detailed characterization of the (optimal) domain index defined in Definition A.2.
>
> **Q3. Architecture and implementation details are missing from the experimental section. This should be provided as supplementary material. Releasing the source code would also be beneficial for reproducibility.**
>
> Thanks for your suggestion. We have added the architecture and implementation details in the Appendix as suggested (Sec. E). We have finished cleaning up the source code and will release it after the paper is accepted to facilitate further research in the community.

---

### Official Review · Reviewer_SAsR · 2022-10-25

**Confidence:** 4
**Correctness:** 3
**Technical Novelty And Significance:** 3
**Empirical Novelty And Significance:** 3
**Recommendation:** 8

**Clarity, Quality, Novelty And Reproducibility:**

The paper is well-written and clearly presented. The problem addressed in this paper is novel and reproducible.

**Strength And Weaknesses:**

- Strength
    1. The paper is well-written and clearly presented.
    2. The proposed method is well-motivated and is theoretically guaranteed.
    3. The idea of automatically inferring the domain index is interesting.
- Weakness
    1. Some of the statements should be further justified. For example, why is leveraging the domain index better than domain identity? Though previous work can somewhat support the statement, it would be better to briefly but explicitly justify it again here.
    2. The domain index is formally defined in section 3.2. However, the definitions and differences between the local index and the global index are not quite clear.  For example, the global index \beta is required to be independent of z, but this is not required for the local index, and why? In addition, the definition does not show the global index is responsible for indicating different domains. Moreover, why we explicitly need both the global index and local index is also not well justified.
    3. The global domain index is inferred based on the local index. Is it necessary to introduce the additional local index? Is it possible to infer the global domain index directly? More analysis and experiments are encouraged for supporting this.

**Summary Of The Paper:**

- To alleviate the reliance on the domain index for domain adaptation, this paper proposes an adversarial variational Bayesian framework that infers domain indices from multi-domain data and provides insights on domain relations to improve domain adaptation performance.
- The main contributions include
    1. analyze and define what is an effective domain index
    2. given the domain identities, infer the domain index as a latent variable from data by developing an adversarial variational Bayesian model.
    3. provide the corresponding theoretical analysis.

**Summary Of The Review:**

In general, the paper is well written and the idea is novel. However, there are still some issues that are required to be addressed as shown in weakness.

---

> ### Author Response · Authors · 2022-11-15
> **[3/3] Thank you for the constructive and encouraging comments**
>
> **Q2.2. For example, the global index $\beta$ is required to be independent of z, but this is not required for the local index, and why?**
>
> **Independence between $z$ and the Global Index $\beta$.** In typical domain adaptation methods, an encoder first takes $x$ as input to produce the representation $z$, and a predictor then takes $z$ as input to predict the label. To improve $z$'s *generalization across different domains*, it is common practice [1, 2, 3] to enforce independence between the domain index $\beta$ and representation $z$ such that *domain-specific information is removed* from $z$. Therefore the assumption/constraint of independence between $\beta$ and $z$ is natural.
>
> **Why Independence between $z$ and the Local Index $u$ is Not Required.** As mentioned in the response to the previous question **Necessity of EMD and MDS.** and our newly added Fig. 8 of Appendix D (where $u$'s in Domain 11 contain two clusters, with one cluster corresponding to positive data and the other corresponding to negative data), we need the local indices $u$ to capture the multi-cluster structure in the data, with each cluster corresponding to local indices for data with the same label $y$. Since $z$ contains label-specific information, if we enforce independence between $z$ and $u$, such label-specific information will be removed from $u$, making it impossible for $u$ to capture the label-specific multi-cluster structure in the data.
>
> We have included the discussion above in the revised paper (Sec. 3.4 and Sec. K in the Appendix).
>
> [1] Yaroslav Ganin, Evgeniya Ustinova, Hana Ajakan, Pascal Germain, Hugo Larochelle, Franccois
> Laviolette, Mario Marchand, and Victor Lempitsky. Domain-adversarial training of neural networks. JMLR 2016.
>
> [2] Hao Wang, Hao He, and Dina Katabi. Continuously indexed domain adaptation. ICML 2020.
>
> [3] Zihao Xu, Guang-He Lee, Yuyang Wang, Hao Wang. Graph-relational domain adaptation. ICLR 2022.
>
>
> **Q3. The global domain index is inferred based on the local index. Is it necessary to introduce the additional local index? Is it possible to infer the global domain index directly? More analysis and experiments are encouraged for supporting this.**
>
> This is a good question and is highly related to our response to Q2.1.
> Local domain indices are necessary for VDI to work. Please kindly refer to the paragraphs **Global versus Local Domain Indices**, **Necessity of Local Domain Indices**, and **Necessity of EMD and MDS** in Q2.1's response above.

---

> ### Author Response · Authors · 2022-11-15
> **[2/3] Thank you for the constructive and encouraging comments**
>
> **Q2.1. The domain index is formally defined in section 3.2. However, the definitions and differences between the local index and the global index are not quite clear...In addition, the definition does not show the global index is responsible for indicating different domains. Moreover, why we explicitly need both the global index and local index is also not well justified.**
>
> This is a good question. Local domain indices are necessary for VDI to work. Below we briefly clarify the difference between global and local domain indices and then discuss the necessity of local indices.
>
> **Global versus Local Domain Indices.** Local domain indices $u$ contain instance-level information, i.e., each data point has a different local domain index $u$; in contrast, global domain indices contain domain-level information, i.e., all data points of the same domain share the same global domain index $\beta$.
>
> **Necessity of Local Domain Indices.** (1) Essentially, local domain indices $u$ serve as extremely low-dimensional summaries of representations $z$; for example, in CompCars experiments the numbers of dimensions for $u$ and $z$ are $4$ and $4096$, respectively. With such low-dimensional $u$, global domain indices can be efficiently inferred from $u$ in the $4$-dimensional space using Eq. (8-11). In contrast, if one only uses global indices without $u$, one would need to compute the Earth Mover's distance (EMD) and multi-dimensional scaling (MDS) in Eq. (8-11) in the $4096$-dimensional space; this is much more computationally expensive, susceptible to overfitting, and dramatically slows down model training. Therefore local domain indices are necessary in VDI. (2) Note that using a smaller number of dimensions for $z$ (e.g., $4$) is not a good choice either, since the predictor takes $z$ as input to predict the label, and using lower-dimensional $z$ harms prediction performance.
>
> **Necessity of EMD and MDS.** It is also worth noting that EMD and MDS are also necessary when inferring global domain indices $\beta$ from local domain indices $u$. The main reason is that $u$'s distribution tends to be *multi-modal*. As an example, we plot the local indices from $3$ domains in *DG-15* in Fig. 8 of Appendix D. We can see that the $u$'s in Domain 11 contain two clusters (i.e. two modes), with one cluster corresponding to positive data and the other corresponding to negative data. This is also the case for both Domain 1 and 0, but the distance between the two clusters gets smaller and smaller. Therefore, directly using the mean of $u$'s as the global domain index $\beta$ does not work because all three domains will have similar means, and consequently similar $\beta$'s. Using EMD and MDS fixes this issue since EMD can naturally compute the distance between two multi-modal distributions. Our preliminary experiments also confirm their necessity; removing EMD and MDS will significantly bring down VDI's performance.
>
> **How the Global Index $\beta$ Indicates Different Domains from the Definition.** One key property that makes sure the global index $\beta$ can distinguish different domains is the second point of Definition 3.1, i.e., *information preservation of $x$*. Specifically, maximizing the mutual information $I(x; u, \beta, z)$ ensures that $\beta$ contains as much information on $x$ as possible (under the constraint that $\beta$ and $z$ is independent); since different domains usually have different distributions over $x$, *information preservation of $x$* helps $\beta$ to distinguish (indicate) different domains. This is also empirically verified by results in Figure 3(d), Figure 5, and Figure 6, where VDI successfully inferred meaningful domain indices $\beta$ of different domains in various datasets. Besides, we also provided an additional figure (Figure 7) of the Appendix to show that the learned global indices $\beta$ highly correlated with the ground-truth indices in the *Circle* dataset.
>
> We have included the discussion above in the revision (Sec. 3.4 and Sec. K in the Appendix).

---

> ### Author Response · Authors · 2022-11-15
> **[1/3] Thank you for the constructive and encouraging comments**
>
> Thank you for your constructive and encouraging comments as well as the insightful questions. We are glad that you find our paper ``"well-written and clearly presented"``, our method ``"well-motivated and theoretically guaranteed"``, and the idea of automatically inferring the domain index ``"interesting"``/``"novel"``. Below we address your questions in detail one by one.
>
> **Q1. Some of the statements should be further justified. For example, why is leveraging the domain index better than domain identity? Though previous work can somewhat support the statement, it would be better to briefly but explicitly justify it again here.**
>
> This is a good suggestion.
>
> **(1)** Compared to domain identities, domain indices can better capture the similarity and complex relations among different domains, thereby better guiding the adaptation process. This is also empirically verified in previous works such as [1, 2]. As a simple example, note that domain identities are one-hot vectors, the distance between any two domain identities is identical; in contrast, domain indices are real-value vectors and therefore contain richer information.
>
> **(2)** More specifically, since the encoder takes domain indices as additional input, domain indices tend to encourage data $x$ of similar domains to go through similar transformations to produce the encoding $z$; consequently, the predicted decision boundaries of similar domains will also be similar.
>
> **(3)** Theoretically, the target domain's error is upper-bounded by three terms, i.e., source error, domain gap, and optimal joint error of the source and target domains [3]. Encouraging similar transformations for similar domains can reduce the joint error term of the upper bound, thereby achieving better performance.
>
> We have added the discussion above in the revised paper (in both the main paper and Sec. J of the Appendix) as suggested.
>
> [1] Hao Wang, Hao He, and Dina Katabi. Continuously indexed domain adaptation. ICML 2020.
>
> [2] Zihao Xu, Guang-He Lee, Yuyang Wang, Hao Wang, et al. Graph-relational domain adaptation. ICLR 2022.
>
> [3] Shai Ben-David, John Blitzer, Koby Crammer, Alex Kulesza, Fernando Pereira, and Jennifer Wortman
> Vaughan. A theory of learning from different domains. Machine Learning, 79(1):151–175, 2010.

---

### Official Review · Reviewer_uSSJ · 2022-10-25

**Confidence:** 4
**Correctness:** 4
**Technical Novelty And Significance:** 3
**Empirical Novelty And Significance:** 3
**Recommendation:** 6

**Clarity, Quality, Novelty And Reproducibility:**

*Clarity
- The order of input variables is inconsistent, such as in Definition 3.1, which is confusing.
- The formats of Eqs.4-7 have problem.
- The description for "(4) Regularization Terms for All Latent Variables" is hard to follow. What kind of regularization is it?
- In the description of the data CompCars, there is no description on what part is source or target.
- It is unclear the proposed method is trained also on the target domain. If so, how do the authors get labels for the target domain?
- It is better to provide some intuition that why the proposed method performed so well in Table 1 since there is a large margin from baselines.
- In the experiment on W->E Level 1, why Source-Only performed the best?
- In figure 6, there is no desctiption for the numbers on points.

*Quality
- Please see the above comments.

*Novelty
- The proposed method seems to be novel.

*Reproducibility
- Code is not available, but the authors provide enough information in the main text.


**Strength And Weaknesses:**

*Strength
- The authors provide a formal definition of domain index from the probabilistic perspective.
- Their theoretical analysis justifies the proposed method.

*Weaknesses
- The assumption of the independence of beta from representation z is not justified well.
- The introduction of the hierarchical modeling of global and local domains is not well supported and motivated. Is there any problem when we only use the global one?
- The paragraph "Inferring Global Domain Indices" in Section 3.4 is hard to follow, and it is a different formulation from Eq.2, which is confusing. Is it impossible to use ELBO simply?

**Summary Of The Paper:**

This paper addresses the domain adaptation problem in VAE.
The authors proposed a method to explicitly estimate the domain index (latent variable) while the domain identity (ID) is known.
The domain index is modeled to be divided into global and local ones and estimated hierarchically. It is proved that the proposed method can optimally obtain them under certain assumptions with the optimization using the proposed loss. The proposed learning procedure imposes that the representation z is learned to be domain invariant.
Experimental results on both synthetic and real-world datasets demonstrate that the proposed method performs much better than existing methods.

**Summary Of The Review:**

The theoretical analysis is solid, but the assumptions behind that are not well justified, such as the independence of beta from representation z and the necessity of beta and u (global and local domain indexes). Clarity is low in general.

================ Update: After the revision in the discussion phase, the assumptions are well justified. Clarity is improved. Thus, I upgraded my score to 6. In the final version, I hope that the authors include a brief description of the efficiency of introducing \u into the main text, which is described in the paragraph "Necessity of Local Domain Indices."

---

> ### Author Response · Authors · 2022-11-15
> **[3/3] Thank you for the constructive and encouraging comments**
>
> **Q6. The description for "(4) Regularization Terms for All Latent Variables" is hard to follow. What kind of regularization is it?**
>
> Eq. 7 includes two KL divergence terms and one entropy term, all of which serve as regularizers to prevent these approximate posteriors of latent variables $u$, $\beta$, and $z$ from overfitting.
>
> For example, the first term implies that the approximate posterior distribution $q(\beta|u)$ (which VDI learns) should be close to the prior distribution $p(\beta)$ (which is set to a standard Gaussian distribution $N(0,I)$). Therefore it is a regularization term for $q(\beta|u)$. Similarly, the second term is also a KL divergence term and therefore ensures that $q_{\phi}(z|x,u, \beta)$ is close to $p_{\theta}(z|x,u, \beta)$.
>
> To better see why KL divergence has a regularization effect, consider the KL divergence $D_{KL}(p||q)$ between two Gaussian distributions $N(\mu_p, \Sigma_p)$ and $N(\mu_q, \Sigma_q)$. We have $D_{KL}(p||q) = \frac{1}{2}[\log\frac{|\Sigma_q|}{|\Sigma_p|} - k + (\mu_p-\mu_q)^T\Sigma_q^{-1}(\mu_p-\mu_q) + tr(\Sigma_q^{-1}\Sigma_p)]$. We can see that minimizing $D_{KL}(p||q)$ tries to bring $\mu_p$ closer to $\mu_q$ and $\Sigma_p$ closer to $\Sigma_q$. Such KL divergence terms often appear in VAE-based models [1].
>
> The last term $-E_{q_{\phi}(u|x)}[\log q_{\phi}(u|x)]$ is the entropy of the approximate posterior distribution $q_{\phi}(u|x)$ (which VDI learns). Note that a Gaussian distribution $N(\mu, \sigma^2)$'s entropy is $\frac{1}{2}ln(2\pi \sigma^2)+\frac{1}{2}$. We can see that maximizing the Gaussian distribution $q_{\phi}(u|x)$'s entropy as a regularization term helps prevent the Gaussian distribution from collapsing to zero variance (i.e., $\sigma^2=0$).
>
> We have included additional clarification in Sec. 3.4 as suggested.
>
> [1] Diederik P Kingma, Max Welling. Auto-encoding variational Bayes. ICLR 2014.
>
> **Q7. In the description of the data CompCars, there is no description on what part is source or target.**
>
> We apologize for missing this detail. We choose the domain with front view (the most common view) and years of manufacture (YOM) 2009 (the earliest YOM) as the source domain and all the others as target domains. We have also revised the paper accordingly (marked in blue).
>
> **Q8. It is unclear the proposed method is trained also on the target domain. If so, how do the authors get labels for the target domain?**
>
> Sorry for the confusion. We follow the typical domain adaptation setting to use labeled data $(x, y)$ from source domains and unlabeled data $x$ from target domains during training. For source-domain data, we use the full ELBO, i.e., Eq (4-7); for target-domain data, we use Eq (4), (6), and (7). We have revised the paper accordingly as suggested to make this clearer.
>
> **Q9. It is better to provide some intuition that why the proposed method performed so well in Table 1 since there is a large margin from baselines.**
>
> This is a good suggestion. Previous work has shown that including domain indices significantly boosts domain adaptation performance [1, 2]; this is because domain indices can better capture the similarity and complex relations among different domains, thereby better guiding the adaptation process. However, domain indices are often unavailable in practice. Our VDI can infer reasonable domain indices (see Figure 3(d), Figure 5, Figure 6, and Figure 7 in the Appendix) from data in an unsupervised manner, and then leverage such inferred domain indices to significantly improve domain adaptation performance. We have added more discussion in Sec. 5.3 of the revised version as suggested.
>
> [1] Hao Wang, Hao He, and Dina Katabi. Continuously indexed domain adaptation. ICML 2020.
>
> [2] Zihao Xu, Guang-He Lee, Yuyang Wang, Hao Wang, et al. Graph-relational domain adaptation. ICLR 2022.
>
> **Q10. In the experiment on W->E Level 1, why Source-Only performed the best?**
>
> Good question. As shown in Figure 4, W->E Level 1 contains the target domains that are the closest to the source domains. Therefore it is relatively easy to adapt from the six source domains on the west coast to these Level-1 target domains. Similar results could be observed in the N->S setting of Table 2, where Source-Only is the strongest baseline for N->S Level-1 domains.
>
> **Q11. In figure 6, there is no description for the numbers on points.**
>
> We are sorry for the confusion. The CompCars dataset consists of 30 domains and these numbers (from $0$ to $29$) on the points in Figure 6 indicate their domain identities (as discrete values). We have included more details in Figure 6 in the revision (marked in blue).

---

> > ### Comment · Reviewer_uSSJ · 2022-11-19
> > **Thank you**
> >
> > Thank you for the effort to address my concerns. I upgraded my score as in the review (see "update" part).

---

> ### Author Response · Authors · 2022-11-15
> **[2/3] Thank you for the constructive and encouraging comments**
>
> **Q3. The paragraph "Inferring Global Domain Indices" in Section 3.4 is hard to follow, and it is a different formulation from Eq.2, which is confusing. Is it impossible to use ELBO simply?**
>
> We are sorry for the confusion. the paragraph "Inferring Global Domain Indices" is actually consistent with Eq. 2. Specifically, Eq. 2 factorizes our variational distribution $q_{\phi}(u,z,\beta|x)$ into three terms, $q_{\phi}(u|x)$, $q_{\phi}(\beta|u)$ and $q_{\phi}(z|x, u,\beta)$. While the terms $q_{\phi}(u|x)$ and $q_{\phi}(z|x, u,\beta)$ are directly parameterized by two neural networks, the remaining term $q_{\phi}(\beta|u)$ is more complicated; this is why we dedicate a separate paragraph, "Inferring Global Domain Indices" in Section 3.4, to describe the details on $q_{\phi}(\beta|u)$.
>
> As mentioned in the paragraph, to aggregate local domain indices $u$ to obtain the global domain indices $\beta$, we need to
>
> **(1)** group local domain indices $u_i$ in each domain $k$ (each domain contains multiple local domain indices),
>
> **(2)** compute pairwise domain distance using the Earth Mover's distance (EMD), where the function $f_{EMD}$ takes as input two sets of $u$'s from two domains and returns their EMD,
>
> **(3)** compute the raw global domain indices using multi-dimensional scaling (MDS), where MDS takes as input the $N$-by-$N$ pairwise domain distance matrix ($N$ is the number of domains) and outputs the raw global domain indices $\beta_k^r$ for each domain $k$, and
>
> **(4)** use a neural network to take $\beta_k^r$ as input and output the final global domain indices $\beta_k$ for each domain.
>
> We can see that $q_{\phi}(\beta|u)$ is much more complicated than directly using a neural network and therefore need a separate description in the paragraph. Our preliminary experiments also confirm the necessity of such complexity; removing EMD and MDS will bring down VDI's performance. Please also refer to the response on **Necessity of EMD and MDS** in the previous question on why such additional complexity is needed. We have included additional clarification in Sec. 3.3 and Sec. 3.4 in the revision as suggested.
>
> **Q4. The order of input variables is inconsistent, such as in Definition 3.1, which is confusing.**
>
> Thanks for mentioning this. We have adjusted the order of input variables such that it is consistent throughout the paper as suggested (for better readability, we did not mark the equations involved in blue).
>
> **Q5. The formats of Eqs.4-7 have problems.**
>
> In Eqs. 4-7, we intentionally group the terms with different meanings in different lines of the long equation (see Eq. 15 on Page 28 of https://www.jmlr.org/papers/volume3/blei03a/blei03a.pdf for a similar format). These four parts of the equation are then explained in the four bullet points below.
>
> Note that Eq. 7 contains three terms with similar meanings; for clarity and convenience when explaining them in Item (4) of the paragraph below Eq. 7, we therefore follow the common practice to group them into the same line with a slightly smaller font size. We are happy to split Eq. 7 into two lines if the reviewer thinks it would be more readable.

---

> ### Author Response · Authors · 2022-11-15
> **[1/3] Thank you for the constructive and encouraging comments**
>
> Thank you for the constructive and encouraging comments as well as the insightful questions. We are glad that you find our proposed method ``"novel"``, our theoretical analysis ``"solid"``, that we ``"provide a formal definition of domain index from the probabilistic perspective"``, that ``"the proposed method performs much better than existing methods"``, and that our ``"theoretical analysis justifies the proposed method"``. Below we address your questions one by one.
>
> **Q1. The assumption of the independence of $\beta$ from representation z is not justified well.**
>
> In typical domain adaptation methods, an encoder first takes $x$ as input to produce the representation $z$, and a predictor then takes $z$ as input to predict the label. To improve $z$'s *generalization across different domains*, it is common practice [1, 2, 3] to enforce independence between the domain index $\beta$ and representation $z$ such that *domain-specific information is removed* from $z$. Therefore the assumption/constraint of independence between $\beta$ and $z$ is natural. We have included the discussion above in the revised paper (Sec. J in the Appendix) as suggested.
>
> [1] Yaroslav Ganin, Evgeniya Ustinova, Hana Ajakan, Pascal Germain, Hugo Larochelle, Franccois
> Laviolette, Mario Marchand, and Victor Lempitsky. Domain-adversarial training of neural networks. JMLR 2016.
>
> [2] Hao Wang, Hao He, and Dina Katabi. Continuously indexed domain adaptation. ICML 2020.
>
> [3] Zihao Xu, Guang-He Lee, Yuyang Wang, Hao Wang. Graph-relational domain adaptation. ICLR 2022.
>
> **Q2. The introduction of the hierarchical modeling of global and local domains is not well supported and motivated. Is there any problem when we only use the global one?**
>
> This is a good question. Local domain indices are necessary for VDI to work. Below we briefly clarify the difference between global and local domain indices and then discuss the necessity of local indices.
>
> **Global versus Local Domain Indices.** Local domain indices $u$ contain instance-level information, i.e., each data point has a different local domain index $u$; in contrast, global domain indices contain domain-level information, i.e., all data points of the same domain share the same global domain index $\beta$.
>
> **Necessity of Local Domain Indices.** (1) Essentially, local domain indices $u$ serve as extremely low-dimensional summaries of representations $z$; for example, in CompCars experiments the numbers of dimensions for $u$ and $z$ are $4$ and $4096$, respectively. With such low-dimensional $u$, global domain indices can be efficiently inferred from $u$ in the $4$-dimensional space using Eq. (8-11). In contrast, if one only uses global indices without $u$, one would need to compute the Earth Mover's distance (EMD) and multi-dimensional scaling (MDS) in Eq. (8-11) in the $4096$-dimensional space; this is much more computationally expensive, susceptible to overfitting, and dramatically slows down model training. Therefore local domain indices are necessary in VDI. (2) Note that using a smaller number of dimensions for $z$ (e.g., $4$) is not a good choice either, since the predictor takes $z$ as input to predict the label, and using lower-dimensional $z$ harms prediction performance.
>
> **Necessity of EMD and MDS.** It is also worth noting that EMD and MDS are also necessary when inferring global domain indices $\beta$ from local domain indices $u$. The main reason is that $u$'s distribution tends to be *multi-modal*. As an example, we plot the local indices from $3$ domains in *DG-15* in Fig. 8 of Appendix D. We can see that the $u$'s in Domain 11 contain two clusters (i.e. two modes), with one cluster corresponding to positive data and the other corresponding to negative data. This is also the case for both Domain 1 and 0, but the distance between the two clusters gets smaller and smaller. Therefore, directly using the mean of $u$'s as the global domain index $\beta$ does not work because all three domains will have similar means, and consequently similar $\beta$'s. Using EMD and MDS fixes this issue since EMD can naturally compute the distance between two multi-modal distributions. Our preliminary experiments also confirm their necessity; removing EMD and MDS will significantly bring down VDI's performance.
>
> We have included the discussion above in the revision (Sec. 3.4 and Sec. K of the Appendix).

---

### Official Review · Reviewer_eAJo · 2022-10-27

**Confidence:** 2
**Correctness:** 4
**Technical Novelty And Significance:** 3
**Empirical Novelty And Significance:** 2
**Recommendation:** 8

**Clarity, Quality, Novelty And Reproducibility:**

Clarity & Quality
I think that the manuscript is well written. I have a minor comment. In Table 2, SENTRY is the best performer for the case “N(24)->S(24): Average of 8 Level-3 domains”.

Novelty & reproducibility
Please see “Strengths and Weaknesses “




**Strength And Weaknesses:**

Strengths.
1.	I think that the problem the authors try to solve is a new (domain adaptation learning with domain indices) and interesting (inference of unknown domain indices) problem.
2.	I think that the manuscript is well written and organized. Although I did not check all the detailed derivations, the theoretical analysis well supports the proposed method.

Weaknesses
1.	The manuscript does not include the analysis of the computational complexities of the proposed method.
2.	Regarding the experiments. I think it would be better to include a table summarizing the data sets, e.g., the input dimension, the number of samples. I also think that it would be better if the previous work, Wang et al. (2020) Xu et al. (2022), had been included in the comparison as all the datasets used in the experiments already include the domain indices. With the same reason, I think that the experiments did not really explain well about what advantages of using the proposed method in practice would be.


**Summary Of The Paper:**

Domain-Indexing Variational Bayes for Domain Adaptation

The manuscript proposes a domain adaption which can infer domain indices (continuous values encoding domain semantics) under the assumptions that domain identities are known but domain indices are not available. The authors first define (local and global) domain indices in the probabilistic viewpoint and then propose an adversarial variational Bayesian framework (the evidence lower bound to learn the approximate posterior distribution for the latent variables and the discrimination term to enforce independence between the global domain indices and the data embeddings) to infer the domain indices. They also provide a theorical analysis which shows that the optimal solution of the proposed objective function is guaranteed to satisfy the three conditions of the domain index they propose to define in the manuscript.


**Summary Of The Review:**

I think that the manuscript has overall high quality.

---

> ### Author Response · Authors · 2022-11-15
> **Thank you for the constructive and encouraging comments**
>
> Thank you for the constructive and encouraging comments. We are glad that you find our research problem ``"new and interesting"``,  our manuscript ``"well written and organized"``, and our theoretical analysis ``"well supports the proposed method"``. Below we address your questions one by one.
>
> **Q1. The manuscript does not include the analysis of the computational complexities of the proposed method.**
>
> The computational complexity of our model is similar to most deep learning models, and therefore our VDI is scalable. The overall complexity is $O(TMP+TN^3)$ where $T$, $M$, $P$, and $N$ are the number of epochs, the number of training data points, and the number of parameters in the neural networks, and the number of domains, respectively (note that $N$ is usually very small). Here $O(TMP)$ is the complexity for optimizing neural network parameters, and $O(TN^3)$ is the complexity for inferring global domain indices, i.e., Eq. (8-11).
>
> Note that VDI's complexity is similar to most deep learning models because it is dominated by the first term $TMP$. To see this, observe that (1) A neural network usually has a large number of parameters $P$ (e.g., in the scale of millions) (2) $N^3$ is much smaller than $MP$ in practice (the number of domains $N$ is the scale of tens in practice). Therefore overall, our VDI scales linearly with $T$, $M$, and $P$. Empirically we found that our VDI's training time is similar to that of other methods such as DANN, MDD, and SENTRY.
>
> **Q2. (1) Regarding the experiments. I think it would be better to include a table summarizing the data sets, e.g., the input dimension, the number of samples.**
>
> Thank you for your suggestion. Following your suggestion, we compile the table summarizing dataset statistics and settings as follows. We have also included the table in Table 4 of the Appendix as suggested.
>
> | Dataset  | Number of Samples | Input Dim        | Toy/Real | Adaptation Task      |
> |:--------:|:-----------------:|:----------------:|:--------:|:--------------------:|
> | Circle   | 3,000             | 2                | Toy      | 2-Way Classification |
> | DG-15    | 1,500             | 2                | Toy      | 2-Way Classification |
> | DG-60    | 6,000             | 2                | Toy      | 2-Way Classification |
> | TPT-48   | 6,912             | 6                | Real     | Regression           |
> | CompCars | 18,735            | 224 $\times$ 224 | Real     | 4-Way Classification |
> | | | |
>
> **Q2. (2) I also think that it would be better if the previous work, Wang et al. (2020) Xu et al. (2022), had been included in the comparison as all the datasets used in the experiments already include the domain indices.**
>
> This is a good question.
>
> **(a)** To evaluate the effectiveness of our VDI, we need to compare VDI's inferred domain indices against domain adaptation datasets that *include ground-truth domain indices*; this makes the datasets of Wang et al. (2020) and Xu et al. (2022) our perfect choice.
>
> **(b)** Wang et al. (2020) and Xu et al. (2022) are not applicable to our setting of inferring domain indices, and direct comparison with our VDI is unfair since Wang et al. (2020) / Xu et al. (2022) and our VDI focus on *different settings* (i.e., the former uses additional information). Specifically, our VDI assumes that domain indices are *not available* and tries to infer these indices, while Wang et al. (2020) and Xu et al. (2022) assume that ground-truth domain indices are *readily available* and use them as additional input to improve performance.
>
> **(c)** It is also worth noting that our VDI is actually complementary to Wang et al. (2020) and Xu et al. (2022) in the case of unknown domain indices, where one could infer domain indices from data and use them as input for Wang et al. (2020) and Xu et al. (2022). This would indeed be interesting future work.
>
> **Q2. (3)  With the same reason, I think that the experiments did not really explain well about what advantages of using the proposed method in practice would be.**
>
> Thanks for mentioning this. We should have emphasized that in practice, domain indices are usually *unknown* (e.g., datasets such as Office-Home and DomainNet); previous domain adaptation methods can only rely on *domain identity* (which contains much less information than domain indices) to align data from different domains and therefore tend to suffer in terms of accuracy. In contrast, our VDI can infer domain indices from data and use these domain indices to improve domain adaptation performance, as shown in Table 2 and 3.
>
> Besides, it is also worth noting that VDI's inferred domain indices can also provide practitioners with meaningful interpretations of the domain adaptation model (see Figure 3(d), Figure 5, and Figure 6).
>
> **Q3. I have a minor comment. In Table 2, SENTRY is the best performer for the case “N(24)->S(24): Average of 8 Level-3 domains”.**
>
> We apologize for the typo and have updated Table 2 accordingly.

---

### Author Response · Authors · 2022-11-15
**Response to all the reviewers and area chairs**

We thank all reviewers for their valuable comments. We are glad that they found the problem we solve ``"new"``/``"interesting"``/``"novel"``/``"well-motivated"`` (eAJo, SAsR, YbH3), our method ``"interesting"``/``"novel"``/``"elegant"`` (eAJo, uSSJ, SAsR, YbH3) and ``"theoretically guaranteed"``/``"well grounded by theoretical results"`` (SAsR, YbH3), our theoretical analysis ``"solid"``/``"well supports the proposed method"`` (uSSJ, eAJo), our experiments ``"demonstrate that the proposed method performs much better than existing methods"``/``"confirm the effectiveness of the method"``/``"interesting"`` (uSSJ, YbH3), and our paper ``"well written"``/``"organized"``/``"presented clearly"`` (eAJo, SAsR, YbH3). Below we address the reviewers’ questions. We have also updated both the main paper and the Appendix (with the changed part marked in blue).

---

### Public Comment · ~Jindong_Wang1 · 2023-02-09
**Suggesting some related work**

Dear authors,

Congrats on this spotlight paper at ICLR!

I just want to suggest some very related work from our side and look forward to your discussion.

First, it turns out that the idea of "domain index" (your work) is conceptually similar to "latent distribution characterization" (our side), which is encouraging because it's good to see there are people with similar ideas:)

Second, we have two papers on this idea. The first one ([Adarnn: Adaptive learning and forecasting of time series](https://arxiv.org/abs/2108.04443), CIKM'21) deals with dynamic distributions in time series, which is a two-stage approach: it first learns the latent distributions (domain index) and then adapts the distributions. The second one ([Out-of-distribution Representation Learning for Time Series Classification](https://arxiv.org/abs/2209.07027), ICLR'23, it was rejected from ICLR'22 with rating 866...) is an end-to-end approach that learns the domain index using adversarial learning.

Finally, to be honest, I've always wanted to use probabilistic model to help model the latent domain distributions but I failed last year... Great to see that you have done it.

I think these work could be discussed or mentioned in you work:) After all, we will also discuss you work in our paper:)

Thanks

Jindong

---

> ### Author Response · Authors · 2023-02-12
> **Reply for the suggestion**
>
> Hi Jindong,
>
> Thank you for your interest in our work and the mentioned references; we will definitely cite them in our revision! Our paper distinguishes between domain identity and domain index, where “domain identity” refers to the one-hot vector that indicates which domain the data point comes from, and “domain index” is a scalar (vector) that reflects the similarity of each domain (more details in the intro). Our VDI tries to infer domain indices given domain identities.
>
> We notice that the concept of “pseudo domain label” or “latent distribution characterization” in your mentioned papers is effectively “domain identity” and therefore is complementary to our VDI. We will be sure to include a similar discussion in our revision. Thank you again for your suggestion and support!

---

### Public Comment · ~Jindong_Wang1 · 2023-02-10
**Further questions**

Hi, I read the paper carefully today and have some questions as below.

1. I still do not understand the role of "local domain index". The experiment section does not show some ablation studies for the improvement brought by local domain index. Additionally, can you provide some real examples of how local domain index exists in real-world applications?
2. How about applying VDI algorithm to public domain adaptation benchmark, e.g., Office-Home, DomainNet, and WILDS? I think that DomainNet is relatively large (.5 M images and 6 domains) and could provide some strong empirical results.
3. Can VDI infer the latent domain index contained in one single domain?

Looking forward to more discussions!

---

> ### Author Response · Authors · 2023-02-12
> **Reply for the questions**
>
> Hi Jingdong,
>
> Thank you for your interest in our work!  Below we address your questions in detail one by one.
>
> **Q1(1): I still do not understand the role of "local domain index". The experiment section does not show some ablation studies for the improvement brought by local domain index.**
>
> This is a good question. Local domain indices are crucial for VDI to work. Without them, VDI could not work effectively, let alone be analyzed in an ablation study.  Below we briefly clarify the difference between global and local domain indices and then discuss the necessity of local indices.
>
> **Global versus Local Domain Indices.** Local domain indices $u$ contain instance-level information, i.e., each data point has a different local domain index $u$; in contrast, global domain indices contain domain-level information, i.e., all data points of the same domain share the same global domain index $\beta$.
>
> **Necessity of Local Domain Indices.** (1) Essentially, local domain indices $u$ serve as extremely low-dimensional summaries of representations $z$; for example, in CompCars experiments the numbers of dimensions for $u$ and $z$ are $4$ and $4096$, respectively. With such low-dimensional $u$, global domain indices can be efficiently inferred from $u$ in the $4$-dimensional space using Eq. (8-11). In contrast, if one only uses global indices without $u$, one would need to compute the Earth Mover's distance (EMD) and multi-dimensional scaling (MDS) in Eq. (8-11) in the $4096$-dimensional space; this is much more computationally expensive, susceptible to overfitting, and dramatically slows down model training. Therefore local domain indices are necessary in VDI. (2) Note that using a smaller number of dimensions for $z$ (e.g., $4$) is not a good choice either, since the predictor takes $z$ as input to predict the label, and using lower-dimensional $z$ harms prediction performance.
>
> **Q1(2):  Additionally, can you provide some real examples of how local domain index exists in real-world applications?**
>
> We can take adaptation on medical diagnosis as an example. In this task, the input is the patients’ physical examination indicators (e.g., blood pressure, total cholesterol), and the output is the diagnosis of a specific disease, such as whether the patient has diabetes. Patients are divided into different age groups (domains) (e.g., 0-16,16-35, 55-70). If we apply VDI to this task, a reasonable local domain index  is the age for each patient, and the global domain index would be the average age of all the patients in the same age group.
>
> **Q2: How about applying the VDI algorithm to public domain adaptation benchmarks, e.g., Office-Home, DomainNet, and WILDS? I think that DomainNet is relatively large (.5 M images and 6 domains) and could provide some strong empirical results.**
>
> This is a good question. To fully demonstrate the advantage of VDI, we prefer to use datasets that have a large number of domains and a clear relationship between each domain. Our 2 real-world datasets, TPT-48 and CompCars, have 48 domains and 30 domains, respectively. Additionally, we understand that states with similar longitude and latitude have similar weather, and cars with similar years of manufacture and viewpoints share similar latent representation. Extending experiments on new datasets, such as DomainNet, would further verify our method’s effectiveness and is a promising direction for future research.
>
> **Q3: Can VDI infer the latent domain index contained in one single domain?**
>
> We assume you are asking if VDI could work when there are only 1 source domain and 1 target domain that share the same data distribution. VDI can still infer their global domain indices, and we expect the two indices to be similar. However, if there is only 1 source domain without any target domains, the global domain index would be meaningless.

---

### Decision · Program_Chairs · 2023-01-20

**Decision:**

Accept: notable-top-25%

**Justification For Why Not Higher Score:**

This paper could potentially be bumped up to an oral.

**Justification For Why Not Lower Score:**

With three overwhelmingly positive reviewers, this is clearly a strong paper.  Also, the topic is important and of relevance to many, so giving the paper at least a spotlight seems justified.

**Metareview: Summary, Strengths And Weaknesses:**

Thanks for your submission to ICLR.  After the rebuttal and discussion phase, all four reviewers were in agreement that this paper is suitable for publication.  The noted weaknesses are mostly minor and many have already been fixed (clarity in the text, additional justifications required).  Strengths included the novelty of the approach, the writing of the paper, and the theoretical analysis to justify the method.

**Note From Pc:**

if the above contains the word "oral" or "spotlight" please see: "oral" presentation means -> notable-top-5% and "spotlight" means -> notable-top-25%. As stated in our emails, we are disassociating presentation type from AC recommendations